# FUND: Density Flow for Sampling Unnormalised Distributions

**Vikas Kanaujia**                                                    *kvikas@iitk.ac.in*
*Department of Electrical Engineering*
*IIT Kanpur*

**Vipul Arora**                                                    *vipul.arora@kuleuven.be*
*Department of Electrical Engineering*
*KU Leuven*

**Reviewed on OpenReview:** *https://openreview.net/forum?id=OO5dDDVcyZ*

## Abstract

Efficient sampling from Boltzmann distributions is central to modelling complex physical systems. Markov Chain Monte Carlo (MCMC) methods suffer from critical slowing down, high autocorrelation, and poor mode-mixing, limiting their scalability. Recent advances, like Boltzmann Generators, offer a promising alternative but remain constrained by costly MCMC-based training, inefficient sampling, and poor ergodicity. We introduce an algorithm for learning Boltzmann distributions that does not require any true samples for training. Our approach draws inspiration from flow matching but departs fundamentally from sample-trajectory matching to distribution-trajectory matching. The algorithm iteratively reshapes the target distribution, using model generated samples to guide learning and ensure comprehensive mode coverage. We validate our method on standard benchmarks, including a 2D Gaussian mixture, Many-Well distributions, and high-dimensional scalar $\phi^4$ theory. The proposed approach not only improves sampling performance and accuracy over traditional MCMC and flow-based baselines but also establishes a new method for sample-free learning of complex physical distributions.

## 1 Introduction

Sampling from unnormalized probability distributions plays a crucial role in understanding and characterizing the behavior of physical systems. In many scientific disciplines, including statistical physics (Gibbs, 2010; Goto et al., 2018), quantum mechanics (Temme et al., 2011), computational chemistry (Noe et al., 2019) and biological sciences (Frauenfelder et al., 1991; Bryngelson et al., 1995), these systems are often represented by Boltzmann distributions, which capture the probabilistic structure of states based on their energy levels.

Sampling such distributions defined as $p(\mathbf{x}) \propto e^{-E(\mathbf{x})}$ with complex energy landscapes in high dimensions pose significant challenges. Traditional samplers such as Markov chain Monte Carlo (MCMC) (Hastings, 1970; Andrieu et al., 2003; Neal, 2011) and Molecular dynamics (MD) (Leimkuhler & Matthews, 2012) proceed via local perturbations, often resulting in slow convergence, high computational cost and poor mode mixing due to a high energy barrier between metastable states. Advances in deep generative models (Song & Ermon, 2019; Song et al., 2020; Wang et al., 2021; Papamakarios et al., 2021; Kobyzev et al., 2021; Lipman et al., 2022; Máté & Fleuret, 2023; Cao et al., 2024; Zhang et al., 2025) have enabled efficient sampling from such distributions and improved the fidelity of generated samples, across several applications in statistical mechanics and biological sciences (Noe et al., 2019; Albergo et al., 2019; Nicoli et al., 2021; Hoogeboom et al., 2022; Watson et al., 2023; Midgley et al., 2023; Akhound-Sadegh et al., 2024; Kanaujia et al., 2024; Zheng et al., 2024; Wang et al., 2024; Njirjak et al., 2024; Ünlü et al., 2025). However, these models typically based on likelihood estimation (Kingma & Welling, 2014; Van den Oord et al., 2016; Papamakarios et al.,

2021; Dinh et al., 2017), adversarial learning (Goodfellow et al., 2014; Arjovsky & Bottou, 2017; Singh et al., 2021), or stochastic regression (Ho et al., 2020; Song et al., 2020) remain highly data-intensive, requiring large training datasets to achieve reliable performance. This limitation is often mitigated by falling back to MCMC or MD methods to generate the training data.

Among these approaches, Normalizing flows (NFs) (Papamakarios et al., 2021), which support exact likelihood evaluation and efficient sample generation, are particularly well suited for approximating Boltzmann distributions. Their flexibility allows training either from samples of the target distribution ($p(\mathbf{x})$) or directly from the underlying energy function. When samples are available, NFs are typically optimized via the forward KL divergence ($\mathcal{KL}(p\|q_\theta)$), which encourages coverage of all modes but tends to overweight low-probability configurations, a well-known mode-covering effect (Nicoli et al., 2023; Kanaujia & Arora, 2025). In the absence of samples, NFs can instead be trained by minimizing the reverse KL divergence ($\mathcal{KL}(q_\theta\|p)$), requiring only energy evaluations up to an additive constant. While this energy-based objective allows learning without data, it biases the model toward a few low-energy regions and the samples generated from the model fail to be ergodic. This phenomenon is referred to as mode collapse (Kanaujia et al., 2024; Midgley et al., 2023). Consequently, despite their advantages, conventional NF training objectives often fail to fully capture the complexity of high-dimensional Boltzmann distributions.

These limitations have prompted growing interest in training strategies that bypass the need for target-distribution samples and rely instead on direct evaluations of the underlying energy function. Such methods aim to build more accurate approximations of the target distribution that can then be used as high-quality proposal distribution. Within this research direction, leading approaches include Flow Annealed Importance Sampling Bootstrap (FAB) (Midgley et al., 2023) and Iterated Denoising Energy Matching (iDEM) (Akhound-Sadegh et al., 2024), both of which employ energy-driven objectives to iteratively enhance the accuracy of the model. FAB couples normalizing flows with annealed importance sampling (AIS), using AIS-generated samples to progressively learn the modelled distribution. While effective in moderate-dimensional settings, this reliance on AIS introduces substantial computational overhead and limits scalability to high dimensions. In contrast, iDEM relies on Monte Carlo (MC) sampling to estimate score. For sampling, it requires solving a reverse SDE, which is computationally expensive. Additionally, iDEM does not yield likelihoods directly: it trains a separate optimal-transport conditional flow-matching model (Tong et al., 2023) on its generated samples, and the reverse ODE of this model is then used to compute negative log-likelihoods (Akhound-Sadegh et al., 2024). This two-stage pipeline is computationally expensive. Moreover, the likelihoods are only approximate and do not ensure bias correction when used with importance weighting or Independent Metropolis Hastings (Kanaujia et al., 2024; Noe et al., 2019).

In this work, we draw inspiration from the framework of Flow Matching (FM) (Lipman et al., 2022) which models the path that a point takes through the evolving probability distribution. This point is realized as a sample that can be seen as carrying a portion of the probability mass. In other words, FM models a way of diffusing or transporting the probability mass from one distribution to another through a continuous transformation using *per sample* paths.

Here, we use this idea of continuous transformations and sample paths to train a tractable sampler $q(\mathbf{x})$, without using any true samples. Let $p_t(\mathbf{x})$ be the continuous density transformation, with $p_1(\mathbf{x}) = p(\mathbf{x})$, the given target distribution, and $p_0(\mathbf{x})$ be a unimodal initial distribution which is easy to sample. Instead of modeling the trajectory of samples with respect to $t$ (as in flow matching), we model the density $q_t$ as a function of $t$ so as to obtain accurate and efficient computation of model density during inference.

Our approach proceeds iteratively, gradually advancing from $t \to 0$ to $t = 1$. At any $t + \delta t$, we generate samples from $q_t$ and re-weight them to learn the distribution $p_{t+\delta t}$ in a supervised way, for example, using weighted FKL or score matching. In addition, we also use sample free objective to learn from the target distribution $p_{t+\delta t}$, for example, RKL. The density trajectories $p_t$ are constructed in such a way that the modes separate gradually. We start with a single mode at $t \to 0$. As $t$ increases, these modes sharpen and separate and our proposed method ensures that $q_t$ learns all the modes. The reweighted samples stabilize the learning as $t$ evolves.

Unlike continuous-time flow matching approaches that learn and integrate a time-dependent velocity field through ODE dynamics, the proposed framework instead learns successively annealed intermediate dis-

tributions, progressively transporting probability mass toward the target distribution without requiring continuous-time integration.

We evaluate our proposed approach on three families of unnormalized distributions. The results demonstrate that our method achieves competitive performance for deep learning–based sample generation from unnormalized target distributions, enabling training without access to true samples that would otherwise require computationally expensive MCMC or molecular-dynamics (MD) simulations.

## 2 Proposed Method

In this section, we outline the proposed approach. We build on the NF framework (Papamakarios et al., 2021) to model complex target distributions by continuously transforming samples drawn from a simple prior into those of the desired distribution through a smooth and invertible mapping. Let $\mathbf{z} \sim q_z(\mathbf{z})$ be sampled from the prior distribution, and let its continuous transformation be $f_t(\mathbf{z}; \theta), t \in [0, 1]$ such that $\mathbf{x} = f_t(\mathbf{z}; \theta)$ should follow $p_t(\mathbf{x})$. The modelled density $q_t$ is given by

$$q_t(\mathbf{x}; \theta) = q_z(f_t^{-1}(\mathbf{x})) |\det[\frac{\partial f_t^{-1}(\mathbf{x})}{\partial \mathbf{x}}]| \tag{1}$$

In contrast to traditional flow-matching methods, which learn the time-dependent trajectories of individual samples, we instead model the evolving density $q_t$ directly as a function of $t$. This formulation enables accurate and computationally efficient evaluation of the model density during inference. To learn the corresponding transformation, we train $f_t$ by minimizing the loss

$$\mathcal{L}(\theta) = \mathbb{E}_{\mathbf{x} \sim p_t}[d(p_t(\mathbf{x}), q_t(\mathbf{x}))] \tag{2}$$

where $\theta$ parameterizes $f_t$, thereby determining $q_t$, and $d$ denotes a distance measure between the densities $p_t$ and $q_t$. For $d$, we can use $f$-divergence or the Fisher divergence. Since samples from $p_t(\mathbf{x})$ are not directly available, we use importance weighting on the samples from $q_{t'}(\mathbf{x}), t' < t$.

$$\mathcal{L}(\theta) = \mathbb{E}_{\mathbf{x} \sim q_{t'}}[w(\mathbf{x})d((p_t(\mathbf{x}), q_t(\mathbf{x})))] \tag{3}$$

where $w(\mathbf{x}) = \frac{p_t(\mathbf{x})}{q_{t'}(\mathbf{x})}$. $q_{t'}$ is learnt inductively starting from $q_0$. Keeping $t'$ closer to $t$ ensures that $w(\mathbf{x})$ remains close to unity, thereby improving numerical stability.

Likelihood- and score-driven objectives typically encourage mode-covering behaviour (Lyu, 2009; Midgley et al., 2023; Kanaujia et al., 2024; Kanaujia & Arora, 2025), often diluting the representation of high-probability regions. To counterbalance this tendency, we introduce an additional reverse KL term in the loss, whose intrinsic mode-seeking behaviour encourages concentration around dominant regions of the distribution. By combining these opposing tendencies, the overall objective balances exploration and concentration, ensuring that all modes are effectively captured.

$$\mathcal{L}_{\mathcal{RKL}}(\theta) = \mathbb{E}_{\mathbf{x} \sim q_t} \log \frac{q_t(\mathbf{x})}{p_t(\mathbf{x})} \tag{4}$$

The final objective function is given by:

$$\mathcal{L}_{final}(\theta) = \lambda_1 \mathcal{L}(\theta) + \lambda_2 \mathcal{L}_{\mathcal{RKL}}(\theta) \tag{5}$$

In this expression, the coefficients $\lambda_1$ and $\lambda_2$ correspond to hyperparameters, the specifics of which are elaborated in Section 2.3. The overall learning procedure proceeds as follows. Starting from a simple unimodal distribution $q_z(\mathbf{z})$, we define a continuous family of densities $\{p_t(\mathbf{x})\}_{t \in (0,1]}$ that transforms smoothly

into the target distribution. The target distribution $p_1(\mathbf{x})$ may be highly multi-modal, but the flow is designed so that its modes appear gradually as $t$ increases (see Figure 2). The model then propagates samples through these intermediate distributions and learns progressively, beginning with lower values of $t$ and advancing toward $t = 1$.

At $t = 0$, the distribution $q$ contains only a single mode. As $t$ increases, additional modes gradually emerge, capturing the increasing complexity of the evolving distribution. Samples drawn from $q_t$ carry information from all these modes, ensuring a smooth transition between distributions over time and effectively preventing mode collapse. In the following, we detail the main components of the proposed method, FUND.

## 2.1 Designing Intermediate Distribution $p_t$

FUND uses a normalising flow to gradually transform the prior as defined in Eq. (1). Rather than modelling individual sample trajectories as in flow matching, FUND models the trajectory of distributions $q_t(x; \theta)$. It learns $q_t$ iteratively using samples from the learnt distribution $q_{t'}$ (with $t' < t$) weighted by importance weights. $t'$ and $t$ are kept sufficiently close to maintain low-variance estimates. We define target trajectory as $p_t(x) \propto p(x)^t$, where $p(x)$ denotes the target distribution. This construction suppresses the peaks and smoothes out the modes for small $t$, substantially reducing the complexity of the distribution and making it easier to learn without missing any mode. As $t$ gradually increases, the modes of $p_t$ sharpen and separate. Figure 2 shows the intermediate target distributions $p_t$ for the 1-D Gaussian mixture, demonstrating the progressive sharpening and separation of modes as $t$ grows.

In our experiments, we approximate the continuous time parameter by discretising the interval $(0, 1]$ into $N$ ordered points $0 < t_1 < t_2 < \cdots < t_N = 1$.

## 2.2 Distance Measure with Importance Weighting

To learn each intermediate distribution, we employ distance measures (refer Eq. (2)) defined with respect to $p_t$, specifically the forward KL divergence and the Fisher divergence. Since direct samples from $p_t$ are not available, we adopt an importance-weighting strategy based on samples drawn from the previously learned distribution $q_{t'}, t' < t$. For sufficiently small increments in $t$, the distribution $q_{t'}$ closely approximates $p_t$, ensuring adequate support overlap and stable importance weights. This enables reliable estimation of both divergence objectives and facilitates the sequential learning of the model across time. Using importance weighting, the forward KL objective can be estimated as

$$\mathcal{L}_{FKL}(\theta) = -\mathbb{E}_{x \sim q_{t'}(x)}[w(x) \log q_t(x)] \tag{6}$$

where $w(x) = \frac{p_t(x)}{q_{t'}(x)}$ denotes the importance weight. Similarly, the Fisher divergence objective (Hyvärinen, 2005) can be written as

$$\mathcal{L}_{\text{score}}(\theta) = \frac{1}{2} \mathbb{E}_{x \sim q_{t'}(x)} \left[ w(x) \left\| \nabla_x \log q_t(x; \theta) - \nabla_x \log p_t(x) \right\|^2 \right]. \tag{7}$$

Here, $\nabla_x \log q_t(x; \theta)$ is computed via automatic differentiation of the model likelihood.

## 2.3 Sequential Modeling Strategy

Several implementation choices exist for FUND. One possible implementation is to construct the model incrementally: A new NF coupling block is appended whenever $t$ is incremented and is trained to minimise the loss in Eq. (5) while the previous blocks are frozen. This design is memory-inefficient and fails to learn all modes of the target distribution as detailed in the Appendix B.

Another implementation is a single-block architecture refined across all time steps. Training proceeds sequentially: the model is first optimised as $q_{t_1}$ with $p_{t_1}$ as the target. At subsequent $t$, the same model is optimised as $q_t$ with $p_t$ as the target. This strategy enables significant memory savings and facilitates incremental learning (Figure 2.2).

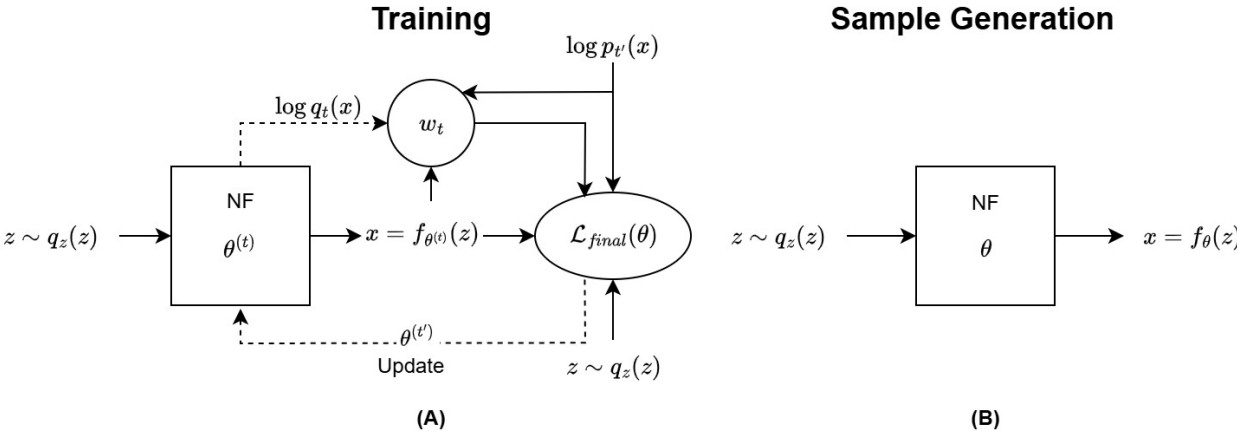

Figure 1: (A) Progressive learning of successive intermediate distributions within a unified flow model using iterative sample reuse. Here $t' > t$, and samples generated from $q_t$ are reused to train $q_{t'}$. Latent samples from $q_z(z)$ are used to compute $\mathcal{L}_{\mathcal{RKL}}(\theta)$, while replay-buffer samples together with $\log q_t(x)$ and $\log p_{t'}(x)$ are used to compute $\mathcal{L}(\theta)$ (B) Illustrates sample generation procedure using the trained flow model.

**Initial training**: Learning begins at $t_1 > 0$, where $\mathcal{L}(\theta)$ is unknown due to the absence of training samples from $p_{t_1}$. We therefore set $\lambda_1 = 0$ and optimize only $\mathcal{L}_{RKL}$. The choice of a small $t_1$ results in a unimodular density $p_{t_1}$ with broadened support, and is easy to learn.

**Sample Buffer:** To ensure efficient reuse of samples during the progression from $t_1$ to $t_N$, we implement a persistent buffer of size $B$ maintained throughout training. At $t_1$, the buffer is populated by transforming prior samples into the intermediate distribution $q_{t_1}$ and saving both the transformed samples and their corresponding log-likelihoods $\log q_{t_1}$. These stored pairs enable the computation of importance weights when training the subsequent distribution $q_{t_2}$.

Reusing the samples of $q_{t'}$ to learn $q_t, t > t'$ can be made efficient and effective by using a prioritised replay buffer (Schaul et al., 2016), which tends to retain samples from different modes and reduces computations. Replay buffers in continual learning are widely used to mitigate catastrophic forgetting by reusing samples from previously learned distributions during subsequent optimization steps (Lesort, 2020; Rolnick et al., 2019). In FUND, the persistent sample buffer continuously reuses samples from earlier intermediate distributions while optimizing later distributions, thereby reinforcing previously learned modes and reducing distributional drift across annealing stages.

Besides, the optimization objective $\mathcal{L}_{final}(\theta)$ integrates $\mathcal{L}_{\mathcal{RKL}}(\theta)$ with $\mathcal{L}(\theta)$ which is either FKL or score-matching supervision. The RKL term uses current transformed samples, whereas the supervision loss leverages samples from previously learned distributions stored in the buffer. As FKL and score-based objectives encourage learning across multiple modes (Kanaujia & Arora, 2025; Kanaujia et al., 2024), the framework empirically maintains stable mode coverage without requiring explicit prioritised replay and mitigates catastrophic forgetting. Thus, we did not need a prioritised replay buffer. We use a uniform replay buffer to store the samples and log-likelihood of $q_{t'}$. While incrementing $t' \to t$, the buffer is updated with samples from $q_{t'}$.

**Hyperparameter annealing**: The training objective incorporates a weighted combination of $\mathcal{L}(\theta)$ and $\mathcal{L}_{RKL}(\theta)$. The optimization begins in a sample-driven regime with $\lambda_1 \gg \lambda_2$, enabling propagation of the modal structure encoded by samples from $q_{t'}$. At each $t$, an annealing schedule gradually reduces $\lambda_1$ and increases $\lambda_2$ based on $-\mathbb{E}_{q_t} \log p_t(x)$, thereby transitioning toward a mode-sharpening phase in which the RKL term enhances mode separation and corrects residual density discrepancies. For more details, refer Appendix C.

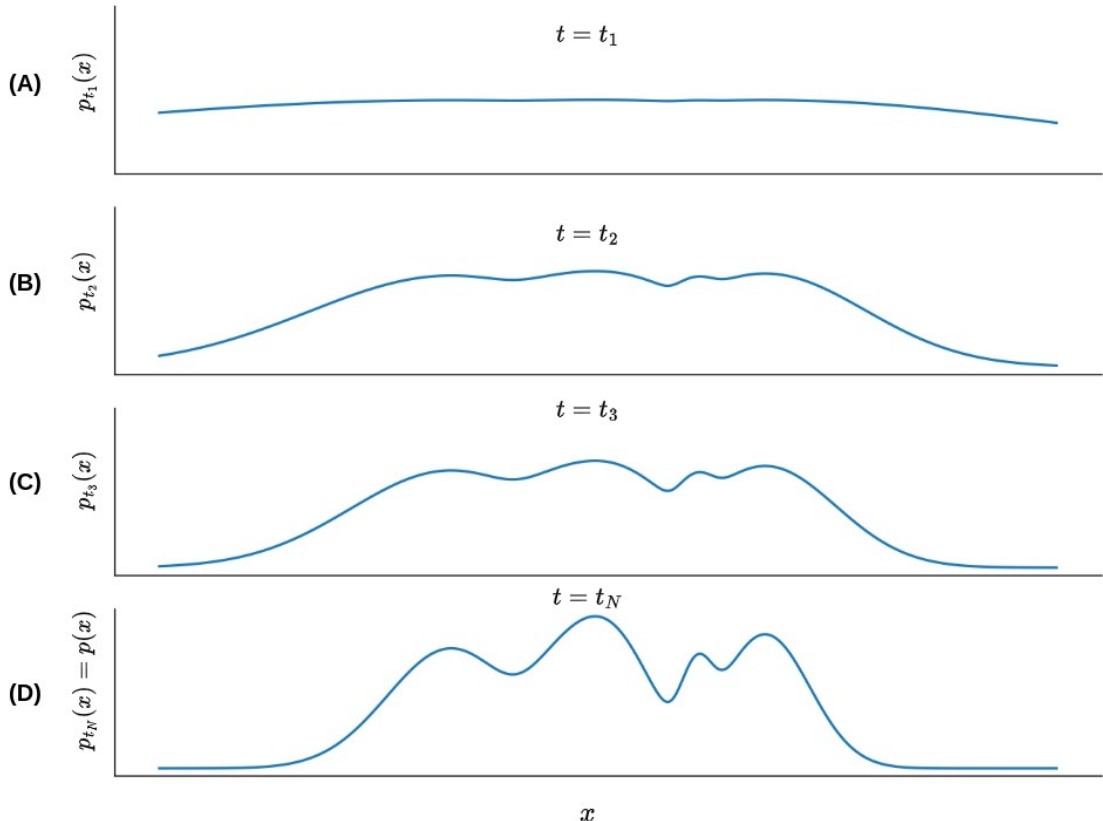

Figure 2: Visualisation of intermediate annealed target distributions $p_t(\mathbf{x}) \propto p(\mathbf{x})^t$ for a 1-D Gaussian mixture model. As $t$ increases, the density gradually transitions from a smoothed, low-contrast form to the fully resolved target distribution. The panel shows 1-D plot for: (A) $t_1 = 0.01$, (B) $t_2 = 0.1$, (C) $t_3 = 0.2$, and (D) $t_N = 1$.

Algorithm 1 outlines the training procedure of FUND, which progressively learns a sequence of intermediate distributions. The method employs buffering to compute importance weights and reuse samples for learning at subsequent time steps. During training, the hyperparameters $\lambda_1$ and $\lambda_2$, which weight the loss components $\mathcal{L}(\theta)$ and $\mathcal{L}_{RKL}(\theta)$, are annealed to ensure stable and accurate convergence at each time step.

## 3 Results

We assess the proposed approach using three metrics: Negative Log-Likelihood (NLL), Reverse Negative Log-Likelihood (RNLL), and Effective Sample Size (ESS).

**NLL** measures how effectively a model fits the observed samples via predicted likelihoods. Lower NLL estimates indicate a model closer to the true distribution.

$$\text{NLL} = -\mathbb{E}_{\mathbf{x} \sim p} \log q(\mathbf{x}) \tag{8}$$

**RNLL**, in conjunction with NLL, provides insight into the model's behaviour across modes. A high NLL accompanied by a low RNLL suggests mode collapse, whereas low NLL and high RNLL reflects broad mode coverage. When both metrics attain low values, the model is well aligned with the target distribution.

$$\text{RNLL} = -\mathbb{E}_{\mathbf{x} \sim q} \log p(\mathbf{x}) \tag{9}$$

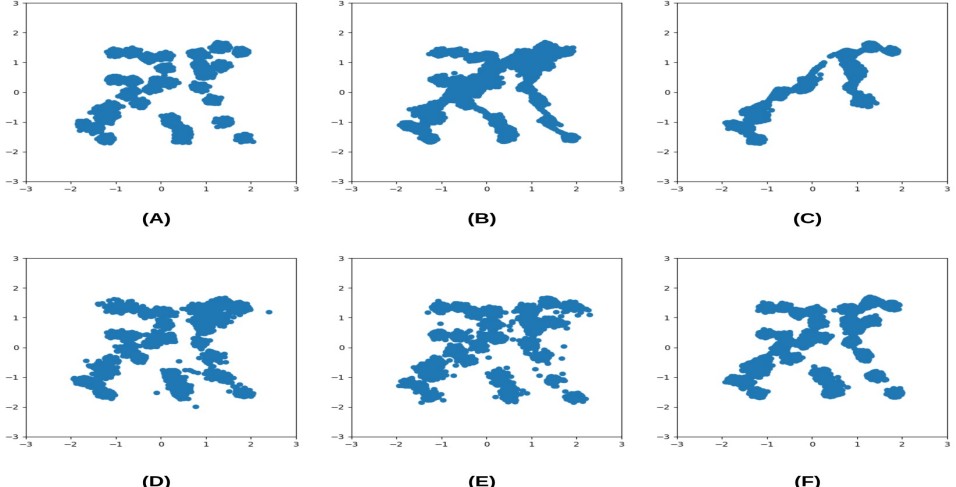

Figure 3: Sample plots for the MOG-40 distribution generated using different methods alongside the corresponding ground-truth samples. These plots illustrate the mode-coverage behavior discussed in the quantitative results and highlight the ability of different methods to preserve the multimodal structure of the target distribution during sampling. (A) Ground-truth samples. (B) FKL. (C) RKL. (D) IDEM. (E) FAB. (F) Proposed FUND(SCORE) method.

**ESS** quantifies how much independent information is effectively represented by a weighted sample set and is defined as

$$\text{ESS} = \frac{(\frac{1}{N} \sum_i p(\mathbf{x}_i)/q(\mathbf{x}_i))^2}{\frac{1}{N} \sum_i (p(\mathbf{x}_i)/q(\mathbf{x}_i))^2} \tag{10}$$

Higher values indicate improved sample diversity and independence.

We compare the proposed framework against four established baselines. (A) **FKL**: An NF model trained by minimising the forward KL divergence (FKL), equivalent to maximum-likelihood estimation and therefore requiring direct samples from the target distribution. (B) **RKL**: An NF model trained using the reverse KL divergence (RKL). (C) **FAB** (Midgley et al., 2023) trains the flow by minimising an $\alpha$-divergence, with expectations approximated using samples generated via AIS. (D) **iDEM** (Akhound-Sadegh et al., 2024) uses stochastic score-matching objective to learn the model. As IDEM does not provide closed-form likelihoods, we approximate its NLL by training an optimal transport conditional flow matching (OT-CFM) model (Tong et al., 2023) on the empirical distribution of iDEM-generated samples.

For the proposed method, we investigate two variants distinguished by the choice of distance measure ($d$) in Eq. (2): **FUND(FKL)**, which employs the forward KL divergence, and **FUND(SCORE)**, which adopts a score-matching based objective. We evaluate the methods across three benchmark Boltzmann distributions as follows:

### 3.1 Mixture of Gaussian (MOG-40)

The MOG-40 benchmark comprises a 2-D mixture of 40 Gaussian components with modes scattered over the range $[-50, 50]$ in both dimensions. This produces a complex, widely separated modal structure that can be clearly visualised from the sample plots. Further information, including the analytical definition of the distribution, is provided in Appendix A.1.

We report the results for the MOG-40 benchmark in Table 1. Both FUND variants perform better than competing methods on MOG-40. FUND(SCORE) provides the best likelihood fit—achieving the lowest NLL—and simultaneously delivers the second-best ESS, signifying strong coverage of the multimodal landscape. FUND(FKL) achieves comparable performance, being slightly worse in NLL and exhibiting slightly

Table 1: Evaluation results for the MOG-40, MW-8, and MW-16 distributions, using the metrics described in Section 3. Best results are highlighted in bold and second best are underlined.

| | MOG-40 | | | MW-8 | | | MW-16 | | |
|---|---|---|---|---|---|---|---|---|---|
| Method | NLL↓ | RNLL↓ | ESS↑ | NLL↓ | RNLL↓ | ESS↑ | NLL↓ | RNLL↓ | ESS↑ |
| FKL | 7.16± 0.02 | 7.87± 0.06 | 0.63± 0.01 | 7.60± 0.05 | -31.60±0.77 | 0.34± 0.02 | 17.14± 0.07 | -48.57± 5.77 | 0.001± 0.004 |
| RKL | 3527.06± 106.44 | 7.07± 0.25 | 0.42± 0.26 | 76.72± 73.84 | -35.40± 0.01 | **0.97**± 0.03 | 604.83± 85.65 | -70.80± 0.06 | **0.83**± 0.06 |
| FAB | 7.14± 0.03 | 8.30± 0.29 | 0.63± 0.06 | 7.00± 0.02 | -33.87± 0.10 | 0.81± 0.01 | 14.28± 0.01 | -66.53± 0.13 | 0.31± 0.02 |
| IDEM | **7.09**± 0.28 | 7.01± 0.30 | 0.42±0.03 | 7.64± 0.09 | -34.86± 0.02 | 0.55± 0.03 | 15.32± 0.43 | -70.08± 0.36 | 0.25± 0.03 |
| FUND(FKL) | 7.11± 0.03 | 7.53± 0.10 | **0.68**± 0.03 | 6.97± 0.01 | -34.25± 0.04 | 0.87± 0.02 | 14.04± 0.05 | -67.95± 0.25 | 0.71± 0.04 |
| FUND(SCORE) | **7.09**± 0.01 | 7.53± 0.01 | 0.67± 0.01 | **6.95**± 0.01 | -34.20± 0.04 | 0.90± 0.01 | **13.93**± 0.02 | -68.35± 0.12 | 0.78± 0.02 |

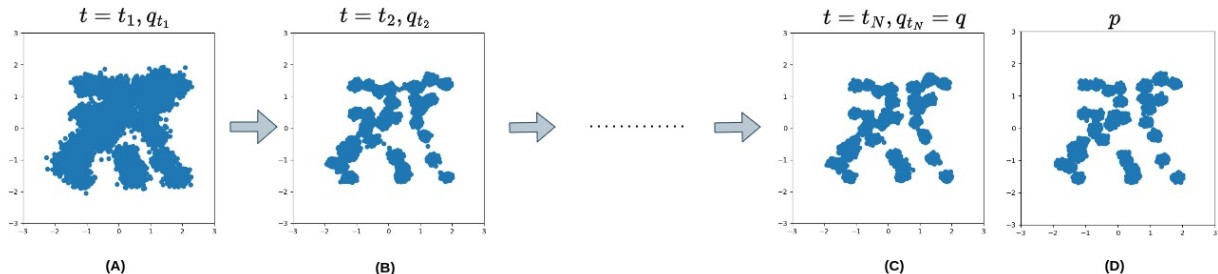

Figure 4: Sample plots of intermediate distributions $q_t$ learned for the MOG-40 distribution, illustrating the gradual transformation induced by the flow model over $t$. Panels show sample plots for distribution : (A) $q_{t_1}$ at $t_1 = 0.01$ (B) $q_{t_2}$ at $t_2 = 0.05$ (C) $q_{t_N}$ at $t_N = 1.0$ and (D) the target distribution $p$.

higher ESS values compared to FUND(SCORE), but still substantially outperforming all baselines. While FKL and FAB obtain moderate NLL values, their ESS scores reveal high correlation among the generated samples. IDEM also achieves the lowest NLL, albeit with a comparatively low ESS value. RKL exhibits severe degradation in NLL, consistent with its strongly mode-seeking behaviour. Although RKL yields the low RNLL value, this result is meaningful only when considered alongside the corresponding NLL. The combination of poor NLL and good RNLL reveals mode collapse, where the model generates samples from only a few modes, artificially inflating RNLL performance. FUND variants achieve better RNLL scores as compared to those of FKL, and FAB, reflecting better alignment with the target distribution. Representative sample visualizations for the MOG-40 distribution generated by different methods are shown in Figure 3, further illustrating the mode-coverage behavior discussed in the quantitative results.

In Figure 4, we illustrate the progression of training using sample plots drawn from the intermediate distributions. At an early time step ($t_1 = 0.01$), the distribution exhibits overlapping modes with a broadened support compared to the target. As training proceeds to $t_2 = 0.05$, the samples from $q_{t_1}$ are used to learn the next intermediate distribution. The sample plot of $q_{t_2}$ shows increasingly sharper and more distinct modes, indicating that the flow gradually refines and preserves the underlying structure.

## 3.2 Many-Well Potential

In sampling tasks, Many-Well Potential (Midgley et al., 2023) represents a benchmark distribution with an energy function characterized by several metastable wells. Sampling from the associated Boltzmann distribution becomes challenging as transitions between wells require overcoming energy barriers, leading to slow mixing and pronounced multimodality. We evaluate our method on 8- and 16-dimensional Many-Well distributions. Appendix A.2 provides the analytical form of the distribution along with other details regarding quantitative evaluation.

Across all baselines, and as demonstrated in Table 1, the proposed method, FUND(SCORE) consistently attains the best well-rounded performance in terms of NLL, RNLL, and ESS. The FUND(FKL) variant performs at par with FUND(SCORE), demonstrating its effectiveness as well.

Table 2: Scalar $\phi^4$ distribution results, with metrics described in Section 3. Best results are highlighted in bold and second best are underlined.

| | $8 \times 8$ | | | $10 \times 10$ | | | $12 \times 12$ | | |
|---|---|---|---|---|---|---|---|---|---|
| Method | NLL↓ | RNLL↓ | ESS↑ | NLL↓ | RNLL↓ | ESS↑ | NLL↓ | RNLL↓ | ESS↑ |
| FKL | 13.40± 0.07 | -9.48± 0.59 | 0.06± 0.01 | 22.01± 0.08 | -17.60± 0.19 | 0.01± 0.00 | 29.89± 0.07 | -32.86± 0.58 | 0.02± 0.004 |
| RKL | 23.18± 0.19 | -18.75± 0.44 | 0.08± 0.05 | 29.01± 1.01 | -29.32± 0.78 | **0.70**± 0.04 | 38.55± 8.29 | -41.32± 0.23 | 0.29± 0.02 |
| FAB | 13.19± 0.21 | -5.14± 0.59 | 0.10± 0.05 | 21.47± 0.22 | -1.05± 0.86 | 0.0001± 0.00 | 31.44± 0.60 | -31.06± 8.35 | 0.09± 0.07 |
| IDEM | 23.76± 1.05 | -23.17± 2.70 | 0.004± 0.004 | 38.13± 2.13 | -25.64± 7.12 | 0.004± 0.004 | 65.84± 4.44 | -15.82± 3.63 | 0.003± 0.005 |
| FUND(FKL) | 12.48± 0.21 | -13.94± 1.52 | 0.14± 0.04 | 19.59± 0.03 | -17.54± 0.83 | 0.14± 0.01 | 28.74± 0.17 | -35.49± 1.29 | 0.33± 0.14 |
| FUND(SCORE) | **12.29**± 0.06 | -14.95± 0.74 | **0.32**± 0.02 | **19.45**± 0.04 | -15.06± 0.62 | 0.15± 0.01 | **28.66**± 0.04 | -36.88± 0.75 | **0.55**± 0.02 |

### 3.3 Scalar $\phi^4$ Theory distribution

Scalar $\phi^4$ theory (Albergo et al., 2019; Singha et al., 2023) is a classical benchmark model in statistical physics and quantum field theory, characterized by a quartic term in its potential. The resulting Boltzmann distribution is highly multimodal, making it a challenging testbed for evaluating sampling algorithms. The corresponding energy function is specified in Appendix A.3. For quantitative analysis, we generate a test set comprising 10,000 samples via HMC. The associated hyperparameter settings are described in the appendix.

Results for the scalar $\phi^4$ model across different lattice sizes are summarised in Table 2. Across all settings, the proposed framework consistently outperforms established baselines. On the $8 \times 8$ lattice, the FUND(SCORE) variant achieves the lowest NLL and the highest ESS, indicating substantially improved sampling efficiency and mode coverage, while FUND(FKL) also surpasses all baselines on both metrics. At $10 \times 10$, the performance gap widens: FUND(SCORE) again attains the best NLL and ESS, with FUND(FKL) closely following. Although RKL yields a moderately high ESS and RNLL, its NLL remains significantly worse, highlighting an imbalance between its mode-seeking behaviour and overall distribution accuracy. In the largest configuration, $12 \times 12$ (144 dimensional distribution), the proposed method maintains strong performance, with FUND(SCORE) achieving the best NLL and ESS, and FUND(FKL) also outperforming all baselines. By contrast, the baselines degrade substantially with increasing dimensionality: FKL, FAB, and IDEM exhibit severe reductions in ESS, while RKL exhibits poor likelihood fit despite moderate ESS values and good RNLL values, learning only a few dominant modes. Overall, the proposed approach, particularly FUND(SCORE), exhibits consistent performance across increasing lattice sizes, thereby demonstrating scalability and superior density estimation capability for high-dimensional Boltzmann distributions arising in lattice field theory.

In Appendix G, we provide a detailed analysis of computational cost, including both training time and inference/sampling time for the scalar $\phi^4$ theory and the MW-16 distribution (Table 9). The results reveal important tradeoffs between computational cost and model performance across the different methods. In particular, FUND(FKL) exhibits training times comparable to FAB while retaining the fast inference characteristics of flow-based models, resulting in a favorable balance between computational cost and sample quality. By contrast, FUND(SCORE) consistently achieves the strongest sampling performance and density estimation accuracy across benchmarks, albeit at a substantially higher training cost, especially for the $12 \times 12$ $\phi^4$ lattice. Although IDEM requires relatively little training time, it incurs significantly higher inference costs than flow-based approaches and produces lower-quality samples. Overall, these results indicate that FUND(FKL) provides an attractive cost–quality tradeoff, whereas FUND(SCORE) offers superior performance at the expense of increased training cost. A comprehensive runtime and computational cost analysis is provided in Appendix G.

## 4 Discussion

In this work, we introduce the idea of gradually shaping the target distribution over time rather than evolving sample trajectories. The flow learns a sequence of intermediate distributions, with each providing samples for the next, resulting in a more accurate approximation of the target distribution. Evaluations on a broad suite of Boltzmann distributions—MOG-40, Many-Well, and scalar $\phi^4$ theory—demonstrate the effectiveness of the proposed method, with results consistently surpassing existing state-of-the-art approaches.

In general, it is preferable to begin with a small initial value of $t_1$, but choosing the small value increases the computational cost. Hence, there is a trade-off. However, we find that for high-dimensional distributions with low multimodality, a larger $t_1$ can be used without affecting performance, offering computational savings. For example, in Tables 1 and 2, we begin with $t_1 = 0.01$ for MOG-40, $t_1 = 0.1$ for Many well and $t_1 = 0.2$ for the $\phi^4$ theory. But a larger $t_1$ may lead to mode collapse in highly multi-modal distributions. For example, in MOG-40, starting with $t_1 = 0.1$ results in missed modes, and this early error propagates to later stages, causing the model to learn only a subset of the modes.

In our setup, $t$ lies in $(0, 1]$, and we discretize this interval into $N$ steps. Larger values of $N$ allow smoother progression between intermediate distributions but at a higher computational cost. Smaller values reduce training time but result in mild performance drops. This trade-off is analysed empirically in Table 5 of the Appendix C.

In the Figure 5, we present the evolution of the test NLL with respect to $t$ for the scalar $\phi^4$ and MW-16 benchmarks, thereby visualising the learning progression over intermediate distributions. Both the variants of FUND consistently exhibit monotonic improvement in NLL. In both cases, the most significant NLL gains occur during the early stages of the transition between successive intermediate distributions, with improvements gradually tapering off as the model converges toward the target distribution at $t = 1$.

Our findings indicate that FUND maintains strong efficiency even for high-dimensional probability distributions. In the scalar $\phi^4$ theory, where dimensionality grows substantially with lattice size, the method delivers consistently strong performance across all evaluation metrics. This contrasts with several baseline approaches, many of which exhibit pronounced degradation as dimensionality increases (Table 2). The stability of FUND under rising complexity highlights its robustness and its ability to model high-dimensional, multimodal Boltzmann distributions without loss of fidelity.

FUND introduces additional sequential optimization stages due to progressive annealing across intermediate distributions. However, this staged optimization improves stability and mode coverage by gradually transporting the model from simpler intermediate densities toward the final multimodal target distribution. As observed in Tables 1 and 2, this sequential distributional learning improves NLL and ESS performance while mitigating mode-collapse behavior. Once training is complete, FUND enables efficient parallel sample generation through a single forward pass of the trained flow model.

In comparison, FAB incurs additional computational overhead from repeated Annealed Importance Sampling (AIS) steps used during optimization to generate training samples across intermediate distributions. This overhead becomes increasingly significant in higher-dimensional settings. In contrast, IDEM has comparatively lower training cost but slower inference, since sample generation requires numerically simulating the reverse-time stochastic differential equation (SDE) using the learned diffusion sampler. MCMC-based methods exhibit the opposite trade-off: they require little or no training cost but incur high inference cost because samples must be generated sequentially through Markov chain transitions, with computational cost scaling directly with the number of generated samples.

Overall, the experiments suggest that FUND(FKL) offers a favorable cost–quality tradeoff, whereas FUND(SCORE) achieves superior sampling performance and accuracy at a higher computational cost. Nevertheless, the sequential annealing strategy yields substantial gains in sample quality and mode coverage while preserving fast parallel inference.

The mode-seeking behavior induced by training with the reverse KL divergence is well documented across variational inference, generative modeling, and Bayesian learning (MacKay, 2003; Bishop & Nasrabadi, 2006). Such behavior often leads to biased approximations, underestimates uncertainty, and insufficient exploration of the global structure of complex distributions—limitations that become especially severe in high-dimensional settings (Blei et al., 2017; Yao et al., 2018). The framework proposed in this work provides an alternative mechanism that mitigates these issues: it reduces the risk of mode collapse, preserves broader support coverage, and enables reliable distribution learning even where reverse KL–based methods typically struggle.

This contribution is also relevant to Bayesian inference, where posterior sampling for complex models is still dominated by classical MCMC techniques such as Metropolis–Hastings, Gibbs sampling, and Hamiltonian-

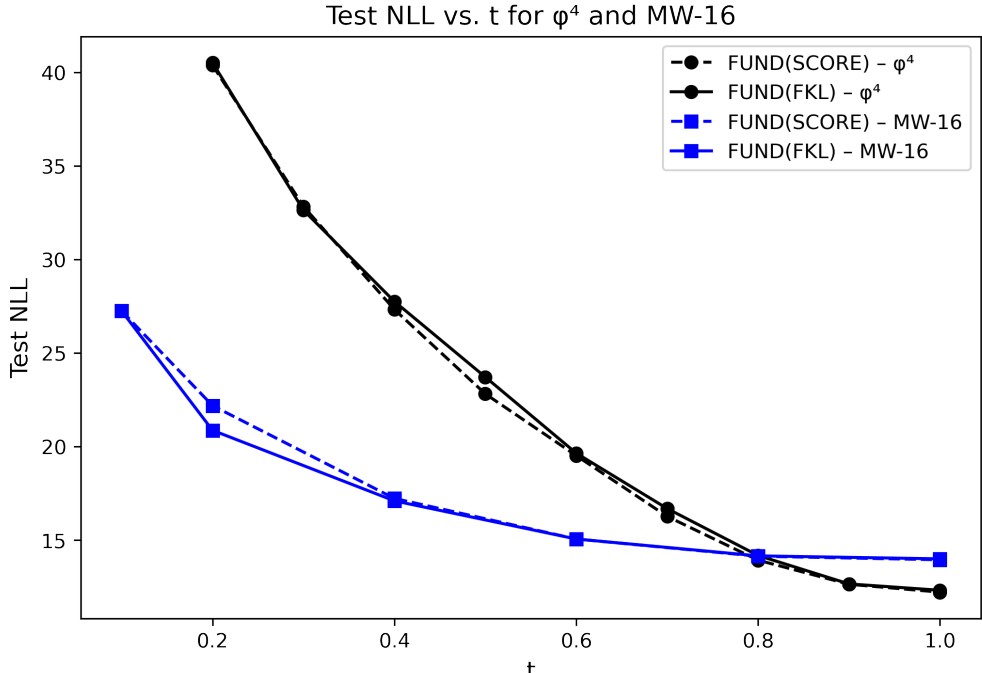

Figure 5: Test negative log-likelihood (NLL) trajectories over $t$ for the scalar $\phi^4$ and MW-16 benchmarks. The curves compare the performance of FUND(SCORE) and FUND(FKL) across intermediate target densities.

based algorithms (Brooks et al., 2011; Neal, 2011). While theoretically principled, these methods often suffer from slow mixing and difficulty in traversing well-separated modes. By supplying a transport-based mechanism capable of capturing the evolving structure of the posterior, FUND offers a pathway toward scalable, amortized, and adaptable sampling.

Finally, the methodology resonates strongly with challenges encountered in lattice field theory, QCD, and gauge-theoretic simulations, where sampling remains highly demanding due to extreme dimensionality, strong correlations, and gauge constraints (Gattringer & Lang, 2009; DeGrand & DeTar, 2006). The corresponding Boltzmann distributions exhibit long autocorrelation times and complex action landscapes, making standard algorithms—such as Hamiltonian Monte Carlo—susceptible to critical slowing down and topological trapping, particularly near the continuum limit (Schaefer et al., 2011). These persistent difficulties underscore the need for more flexible sampling strategies that maintain gauge fidelity while improving exploration efficiency. Our approach directly addresses these requirements and therefore holds significant potential for future applications in lattice and gauge-theoretic contexts.

The primary constraint of the proposed framework lies in its sequential learning procedure, which requires the model to progress through intermediate distributions one stage at a time. While computational cost can be reduced by using fewer intermediates, this must be balanced against the need to maintain smooth structural evolution of the distribution; overly coarse discretization may disrupt the transition dynamics and hinder accurate learning.

## 5 Conclusion

This work presents FUND, a novel learning framework for training NFs on unnormalised target distributions without requiring access to true target samples. Unlike existing sample-driven approaches, the proposed method progressively shapes the model distribution through a sequence of intermediate distributions of

increasing complexity. By using importance weighted samples from the intermediate distributions, FUND achieves stable learning, comprehensive mode coverage, and tractable likelihood evaluation.

The effectiveness of the proposed approach is demonstrated across several challenging benchmarks, including MOG-40, Many-Well, and scalar $\phi^4$ theory. The results show that FUND consistently outperforms state-of-the-art approaches, particularly in high-dimensional and highly multimodal settings. Computational cost analysis further reveals important tradeoffs between model quality and runtime characteristics, with FUND(FKL) providing a favorable balance between training cost, inference speed, and sample quality, while FUND(SCORE) achieves the higher sample quality with superior sampling performance and accuracy, albeit at the expense of increased training cost. Overall, this work establishes FUND as a robust and scalable framework for learning complex unnormalised distributions without reliance on target samples.

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

# A    Distributions

In this section, we briefly discuss the distributions that we modelled.

## A.1    Mixture of Gaussian (MOG-40)

It is defined as a mixture of 40 Gaussian components in $\mathbb{R}^2$. The density function is given by

$$p(\mathbf{x}) = \sum_{i=1}^{N} a_i \, \mathcal{N}(\mathbf{x}; \boldsymbol{\mu}_i, \Sigma_i), \tag{11}$$

with mixture weights $a_i > 0$, means $\boldsymbol{\mu}_i \in \mathbb{R}^2$, and covariance matrices $\Sigma_i \in \mathbb{R}^{2 \times 2}$. For the MOG-40 specification, we choose $a_i = 1/40$, sample $\boldsymbol{\mu}_i \sim \mathcal{U}(-40, 40)$, and set $\Sigma_i = I$. This produces a structured multimodal distribution suitable for evaluation. We generate a test set of 5,000 samples following Midgley et al. (2023) with a fixed seed of 2000. In addition, the FKL baseline requires data for model training. We therefore also generate 10,000 samples for training, together with 5,000 samples for validation.

## A.2    Many-Well Distribution

It is a synthetic distribution constructed as the product of $d/2$ independent copies of the two-dimensional double-well distribution (Noe et al., 2019), where $d$ denotes the dimensionality of the space. This construction produces an exponentially large number of modes $(2^{d/2})$, making the distribution challenging for density estimation and sampling tasks. Its energy function is given by

$$H(\mathbf{x}) = \sum_{i=1}^{d/2} \left( x_{2i-1}^4 - 6x_{2i-1}^2 - 0.5x_{2i-1} + 0.5x_{2i}^2 \right), \quad \mathbf{x} \in \mathbb{R}^d \tag{12}$$

To study performance across increasing multimodality, we experiment with both MW-8 and MW-16. Using rejection sampling, as in Midgley et al. (2023), we generate 10,000 test samples, along with an additional 10,000 training and 10,000 validation samples for the FKL baseline for each distribution.

## A.3    Scalar $\phi^4$ theory distribution

It is a fundamental lattice field theory model widely used in computational physics to study interacting scalar fields, renormalisation, symmetry breaking, and critical phenomena. A real scalar field $\mathbf{x}$ defined on

Table 3: Comparison between incremental modelling and single block modelling on MOG-40.

| Modelling strategy | NLL($\downarrow$) |
|---|---|
| Incremental Modelling | 7.69 |
| Single Block Modelling | **7.07** |

a two-dimensional square lattice with $d$ sites is characterised by the energy function

$$H(\mathbf{x}) = \sum_{l=1}^{d} \left( \lambda x_l^4 + m^2 x_l^2 + 2 \sum_{l' \in n(l)} (x_l^2 - x_l x_{l'}) \right) \tag{13}$$

The set $n(l)$ identifies the two nearest neighbours of each site $l$. Here, $\lambda$ parameterises the quartic self-interaction, while $m$ specifies the mass term. This is commonly used as a benchmark in numerical simulations and sampling studies. To study the proposed method across different system scales, we conduct experiments on three lattice sizes: $8 \times 8$, $10 \times 10$, and $12 \times 12$. Following the setup in Singha et al. (2023), samples are drawn using HMC (Neal, 2011) with parameters $\lambda = 4$ and $m^2 = -4$. The integrator uses a step size of 0.05 and 20 leapfrog steps per trajectory. We discard the first 1,000 samples on account of thermalization. A total of 10,000 samples are generated for testing, and for the FKL baseline, we additionally generate 10,000 training and 10,000 validation samples.

## B  Model details

In this section, we provide an overview of the model architecture used in our experiments. As discussed under modelling strategy in Section 2, we first explored an incremental modelling strategy. In this setup, with the increment of the time parameter $t$, we add new coupling blocks atop the previously trained ones, allowing the model to progressively learn the intermediate distributions. During training, the model encounters difficulties at intermediate time instants, leading to missed modes and results in degradation of test NLL. Furthermore, the incremental construction is inherently memory-inefficient. Besides, each added coupling block required training from scratch, while previously learned blocks remained frozen, limiting overall training efficiency. Hence, we adopt another efficient implementation that employs a single-block model (see in Figure 2.2) refined across all time steps. Starting with the optimisation of $q_{t_1}$ toward $p_{t_1}$, the same block is subsequently updated for each later $t$. This approach greatly reduces memory usage and enables incremental learning. Our experiments on the MOG-40 distribution also support this. The incremental modelling strategy produces a considerably higher NLL, while single block implementation delivers improved performance, as summarised in Table 3. Throughout the paper, we use the single block implementation for FUND.

We implement the model using the RNVP architecture as in Dinh et al. (2017) using Affine coupling blocks. For the MOG-40 and Many-Well distributions defined in $\mathbb{R}^d$ space, the affine coupling block is realized using fully connected (dense) networks. Each coupling block consists of two dense layers, each comprising $D$ hidden units equipped with ReLU activation functions. The network then branches into two output layers corresponding to the scale $s$ and translation $t$. The scale layer uses a `tanh` activation to output the scaling factors, while the translation layer provides a linear output.

For the scalar $\phi^4$ distribution, the field $x$ is defined on a 2D square lattice with $d = N_1 \times N_1$ sites. The affine coupling block is implemented using convolutional layers: two convolutional layers with $F$ filters of size $3 \times 3$, periodic padding, and ReLU activation. The output then branches into two layers—one producing the scale $s$ and the other the translation $t$. Each layer uses a single $3 \times 3$ filter, with tanh activation applied to the scale and a linear activation applied to the translation. Table 4 summarizes the affine coupling configurations used across all distributions, including the number of coupling blocks and the corresponding filter count $F$ (for convolutional layers) and neuron count $D$ (for dense layers).

For MOG-40, MW-8, and MW-16, we adopt a batch size of 512, whereas for the scalar $\phi^4$ theory distribution we use a batch size of 256. All models are trained with the Adam optimizer and an initial learning rate of

Table 4: Summary of the NF architecture configurations used across different distributions.

| Distribution | # Affine Coupling Blocks | F or D |
|---|---|---|
| MOG-40 | 16 | 320 |
| MW-8 | 12 | 512 |
| MW-16 | 12 | 512 |
| Scalar $\phi^4(8 \times 8)$ | 10 | 256 |
| Scalar $\phi^4(10 \times 10)$ | 12 | 256 |
| Scalar $\phi^4(12 \times 12)$ | 12 | 256 |

$1 \times 10^{-5}$. The learning rate is lowered during the final stages, specifically for the last two time instants, $t_{N-1}$ and $t_N$.

For baseline models FAB and IDEM, we primarily followed the architectures, optimizer settings, and training procedures recommended in the respective original papers and official repositories. In addition, we performed limited hyperparameter tuning to ensure stable convergence and competitive performance on our benchmark distributions.

Since both FAB and FUND are flow-based methods, we kept the normalizing-flow (NF) architectures identical wherever possible to ensure a fair comparison. In particular, the NF configurations for FUND and FAB were matched for the scalar $\phi^4$ experiments. Complete architectural details are provided in Table 4 of Appendix B.

For MOG-40, MW-8 and MW-16, we used the recommended settings reported in the original FAB paper and official implementation repository. In FAB experiments, hyperparameter tuning primarily involved the number of intermediate annealing distributions and the number of MCMC transition steps. The key hyperparameters used in the FAB experiments are summarized here. For MOG-40, we used Metropolis transitions with 1 intermediate distribution, 1 transition step per iteration, a replay buffer size of 12,800, and 20 million forward passes. For MW-8 and MW-16, we used HMC transitions with 4 intermediate distributions, 5 transition steps, a replay buffer size of 12,800, and 50 million forward passes. For the $\phi^4$ experiments, HMC was used as the transition operator with 4 intermediate distributions and 5 transition steps. The replay buffer size was set to 12,800 for the $8 \times 8$ lattice and 25,600 for the $10 \times 10$ and $12 \times 12$ lattices. The total number of forward passes was 50 million for the $8 \times 8$ lattice and 100 million for the $10 \times 10$ and $12 \times 12$ lattices. The learning rate was fixed at $1 \times 10^{-4}$ across all experiments

Similarly, for IDEM, we largely followed the official implementation and only performed limited tuning for the $\phi^4$ experiments, including learning rate, batch size, and model depth/width within reasonable ranges. We used an MLP architecture consisting of 4 hidden layers with 512 neurons per layer and sinusoidal embeddings of dimension 512. All models were trained for 1000 epochs using a replay buffer with a maximum capacity of 10,000 samples. A geometric noise schedule of the form $\sigma(t) = \sigma_{\min}(\sigma_{\max}/\sigma_{\min})^t$ was employed across all experiments with $\sigma_{\min} = 10^{-5}$. For MOG-40, we used $\sigma_{\max} = 1.0$, learning rate $5 \times 10^{-4}$, and $K = 500$ samples for computing the regression target $S_K$, with the target norm clipped to 70. For MW-8 and MW-16, we used $\sigma_{\max} = 3.0$, learning rate $10^{-3}$, and $K = 1000$ samples, with the regression target norm clipped to 20. For the $\phi^4$ benchmarks ($8 \times 8$, $10 \times 10$, and $12 \times 12$), we used $\sigma_{\max} = 1.5$, learning rate $10^{-4}$, and $K = 1000$ samples, with the regression target norm clipped to 20. The remaining hyperparameter settings were adopted from the original repository.

For the remaining baselines, namely FKL and RKL, we used the same RealNVP architecture as employed in FUND to ensure a fair comparison across flow-based models. We did not perform exhaustive large-scale hyperparameter searches for any method. All methods were trained under comparable computational budgets and hardware settings to ensure a fair empirical comparison.

## C  Hyperparameter Selection and Tuning

The proposed approach introduces several hyperparameters that shape its learning dynamics, including the number of intermediate distributions that determine the transition path, the initial timestep that affects early-mode coverage, the buffer size used for storing training samples, and the weights that govern how the two loss terms contribute at different stages of optimization (Eq. (5)).

The selection of $N$ is particularly impactful and plays an important role in shaping of intermediate distributions. It determines how finely the $t \in (0,1]$ is discretized. Larger $N$ allows for smoother interpolation, albeit at increased computational cost. Smaller values improve efficiency but lead to reduced performance. In Table 5, we report the impact of different values of $N$ on the $8 \times 8$ scalar $\phi^4$ theory distribution. As $N$ increases, performance consistently improves across metrics, highlighting that smoother transitions between intermediates enhance both fidelity and sampling efficiency.

Table 5: Results for the scalar $\phi^4$ model distribution on an $8 \times 8$ lattice, illustrating the effect of varying the number of intermediate distributions $N$ on model performance.

| N | NLL($\downarrow$) | RNLL($\downarrow$) | ESS($\uparrow$) |
|---|---|---|---|
| 4 | 15.04 | 3.25 | 0.02 |
| 6 | 13.13 | -9.75 | 0.02 |
| 9 | **12.34** | **-15.68** | **0.15** |

As discussed in Section 4, the initialization of $t$ is central to ensuring that the model captures all the modes of the target distribution before progressing through the intermediate stages. Although very small starting values are often ideal, our empirical findings demonstrate that in high-dimensional scenarios, starting from a moderately larger $t_1$ does not degrade performance. This provides a practical advantage by lowering computational requirements while maintaining reliable distributional learning. We tune this parameter via grid search to ensure stable learning across all distributions. Table 6 provides the number of intermediate distributions $N$ selected and the specific discretized $t$ values used for each target distribution during training.

Table 6: Intermediate distribution parameters $N$ and $t$-schedules used in training.

| Target Distribution | N | t |
|---|---|---|
| MOG-40 | 7 | {0.01,0.05,0.1,0.25,0.50,0.75,1.0} |
| MW-8 | 6 | {0.1,0.2,0.4,0.6,0.8,1.0} |
| MW-16 | 6 | {0.1,0.2,0.4,0.6,0.8,1.00} |
| Scalar $\phi^4(8 \times 8)$ | 9 | {0.2,0.3,0.4,0.5,0.6,0.7,0.8,0.9,1.0} |
| Scalar $\phi^4(10 \times 10)$ | 9 | {0.2,0.3,0.4,0.5,0.6,0.7,0.8,0.9,1.0} |
| Scalar $\phi^4(12 \times 12)$ | 9 | {0.2,0.3,0.4,0.5,0.6,0.7,0.8,0.9,1.0} |

Since training at stage $t$ depends on samples generated at the previous intermediate distribution as detailed in Section 2, the size of the buffer $B$ directly influences the quality of the learned transitions. A sufficiently large buffer helps maintain broad support and prevents the loss of modes, particularly in high-dimensional or strongly multimodal settings. While smaller buffers reduce computational overhead but may compromise stability if mode structure is complex. For the experiments conducted in this work, we set the buffer size to 10,000 for MOG-40, MW-8, and MW-16. For the $\phi^4$ model, we used 10,000 samples for the $8 \times 8$ lattice and increased the buffer to 25,000 for the $10 \times 10$ and $12 \times 12$ cases to preserve sample diversity.

Within our framework, training proceeds using a composite loss involving $\mathcal{L}(\theta)$ and $\mathcal{L}_{\mathrm{RKL}}(\theta)$, weighted by the hyperparameters $\lambda_1$ and $\lambda_2$. At the first intermediate level $t = t_1$, samples are unavailable, necessitating the use of the reverse KL term only by setting $\lambda_1 = 0$ and $\lambda_2 = 1$. For all subsequent $t$, optimization leverages samples generated at the previous time step, with $\lambda_1$ initially assigned a larger value to maintain high fidelity to the previously learned mode structure. This configuration is retained until improvements in

the monitored reverse NLL objective begin to plateau. Beyond this point, an annealing schedule gradually lowers $\lambda_1$ while increasing $\lambda_2$, thereby transitioning the algorithm toward a refinement phase in which the reverse KL term promotes sharper mode resolution and fine-scale correction of residual density mismatches.

## D Sensitivity Analysis of the Loss Annealing Schedule

Training in FUND proceeds using the composite objective

$$\mathcal{L}_{\{\}\backslash\neg\updownarrow}(\theta) = \lambda_1 \mathcal{L}(\theta) + \lambda_2 \mathcal{L}_{\mathcal{RKL}}(\theta), \tag{14}$$

where $\mathcal{L}(\theta)$ denotes either the FKL or score-matching objective, and $\mathcal{L}_{\mathrm{RKL}}(\theta)$ denotes the reverse KL regularization term.

At the first annealing stage $t = t_1$, replay-buffer samples are unavailable. Consequently, optimization initially relies only on the reverse KL objective by setting

$$\lambda_1 = 0, \qquad \lambda_2 = 1. \tag{15}$$

For all subsequent annealing stages, optimization additionally leverages replay-buffer samples generated at the previous timestep. In this phase, $\lambda_1$ is initially assigned a relatively larger value to preserve previously learned mode structure and maintain continuity along the annealed trajectory. As training progresses and the reverse NLL objective begins to plateau, the optimization gradually transitions toward stronger reverse-KL-based refinement by decreasing $\lambda_1$ and increasing $\lambda_2$.

To study the sensitivity of FUND to this annealing strategy, we conducted an ablation study on the MW-8 distribution using the FUND(FKL) variant under four different annealing schedules.

**Setting 1**

$$(1,1) \to (0.5,1) \to (0.25,1) \to (0.1,1) \to (0.01,1) \tag{16}$$

Here, $\lambda_1$ is annealed down from 1 to 0.01, while $\lambda_2$ remains fixed at 1.

**Setting 2**

$$(1,0.1) \to (1,0.25) \to (1,0.5) \to (1,1) \tag{17}$$

Here, $\lambda_1$ remains fixed at 1, while $\lambda_2$ is annealed upward from 0.1 to 1.

**Setting 3**

$$(1,0.1) \to (1,0.5) \to (0.5,1) \to (0.1,1) \tag{18}$$

In this setting, $\lambda_1$ is annealed downward while $\lambda_2$ is annealed upward over four stages.

**Setting 4**

$$(1,0.1) \to (1,0.25) \to (1,0.5) \to (1,1) \to (0.5,1) \to (0.25,1) \to (0.1,1) \to (0.01,1) \tag{19}$$

This schedule is similar to Setting 3 but performs the annealing more gradually over eight stages.

Across all settings, we observed that FUND remains reasonably stable provided that:

- the supervision objective $\mathcal{L}(\theta)$ dominates during the early stages of optimization, and

- the transition between $\mathcal{L}(\theta)$ and $\mathcal{L}_{\mathrm{RKL}}(\theta)$ occurs gradually rather than abruptly.

Table 7: Sensitivity analysis of the annealing schedule for $(\lambda_1, \lambda_2)$ on the MW-8 distribution using FUND(FKL).

| Setting | Annealing Schedule $(\lambda_1, \lambda_2)$ | NLL ↓ | RNLL ↓ | ESS ↑ |
|---|---|---|---|---|
| Setting 1 | $(1,1) \to (0.5,1) \to (0.25,1) \to (0.1,1) \to (0.01,1)$ | 7.07 | -33.60 | 0.79 |
| Setting 2 | $(1,0.1) \to (1,0.25) \to (1,0.5) \to (1,1)$ | 7.05 | -33.46 | 0.77 |
| Setting 3 | $(1,0.1) \to (1,0.5) \to (0.5,1) \to (0.1,1)$ | 7.03 | -33.62 | 0.82 |
| Setting 4 | $(1,0.1) \to (1,0.25) \to (1,0.5) \to (1,1) \to (0.5,1) \to (0.25,1) \to (0.1,1) \to (0.01,1)$ | **6.96** | **-34.30** | **0.88** |

Table 8: Sensitivity analysis of the initial annealing parameter $t_1$ on the MOG-40 benchmark.

| $t_1$ | NLL ($\downarrow$) | Number of Modes Recovered |
|---|---|---|
| 0.010 | 7.07 | 40 |
| 0.025 | 20.03 | 33 |
| 0.050 | 53.20 | 20 |
| 0.075 | 990.99 | 15 |
| 0.100 | 1327.82 | 12 |

The overall performance trends remained qualitatively consistent across all schedules. However, keeping either $\lambda_1$ or $\lambda_2$ effectively fixed throughout training (Settings 1 and 2) generally resulted in inferior performance. Gradual annealing of both coefficients yielded the best empirical performance (Setting 4), suggesting that a smooth transition between mode preservation and reverse-KL-based refinement is beneficial. However, this schedule also incurs higher computational cost due to the increased number of optimization stages.

In contrast, Setting 3 provides a favorable trade-off between computational efficiency and performance, achieving results close to Setting 4 while requiring fewer annealing stages.

# E   Sensitivity Analysis of Initial Annealing Parameter $t_1$

The choice of the initial annealing parameter $t_1$ plays an important role in the stability of the method and the preservation of mode coverage during the early stages of training.

As discussed in the main manuscript, choosing a sufficiently small $t_1$ smooths the target landscape and connects otherwise isolated modes, thereby simplifying the initial optimization problem. However, when $t_1$ is chosen too large, the initial annealed distribution remains highly multimodal, increasing the mode-seeking tendency of reverse-KL optimization and potentially leading to mode collapse.

To better characterize this behavior, we conducted additional sensitivity experiments on the MOG-40 benchmark by varying $t_1$ and measuring both the number of recovered modes and the corresponding NLL values. The quantitative results are summarized in Table 8.

Our observations indicate a clear trend: larger values of $t_1$ lead to severe mode collapse, whereas sufficiently small values of $t_1$ significantly improve stability and preserve the modal structure of the target distribution. In particular, we observed noticeable mode collapse at $t_1 = 0.1$, while using $t_1 = 0.01$ successfully preserved all major modes of the MOG-40 distribution.

Figure 6 further illustrates the effect of $t_1$ on sample quality and mode coverage. The results visually demonstrate that smaller initial annealing values improve support coverage and help maintain the multimodal structure of the target distribution throughout optimization.

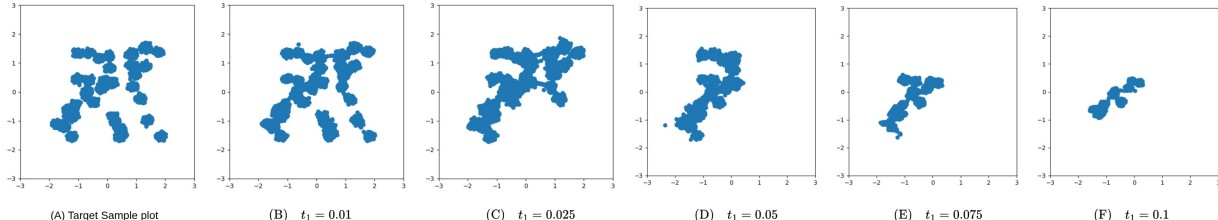

(A) Target Sample plot  (B) $t_1 = 0.01$  (C) $t_1 = 0.025$  (D) $t_1 = 0.05$  (E) $t_1 = 0.075$  (F) $t_1 = 0.1$

Figure 6: Effect of the initial annealing parameter $t_1$ on MOG-40. The figure shows sample plots of (A) the target distribution and (B)–(F) generated samples for $t_1 = 0.01$, 0.025, 0.05, 0.075, and 0.1, respectively. Larger $t_1$ values lead to mode collapse.

## F  Practical Guidelines for Hyperparameter Selection

FUND involves several hyperparameters that influence optimization stability, mode coverage, and computational cost. In this section, we summarize practical guidelines and empirical heuristics for selecting these hyperparameters when applying the method to new target distributions.

### F.1  Choice of Initial Annealing Parameter $t_1$

The most important hyperparameter is the initial annealing parameter $t_1$, since it controls the smoothness of the first intermediate distribution:

$$p_{t_1}(x) \propto p(x)^{t_1}.$$

In general:

- highly multimodal distributions with sharp energy barriers benefit from smaller values of $t_1$,

- whereas simpler or weakly multimodal targets with high dimensionality can tolerate relatively larger values.

Intuitively, $t_1$ should be chosen sufficiently small so that the initial annealed distribution exhibits overlap across modes and can be learned without severe mode collapse. If $t_1$ is too large, the initial optimization problem may remain highly multimodal, increasing the mode-seeking tendency of reverse-KL optimization.

The sensitivity analysis presented in Appendix E demonstrates this behavior empirically on the MOG-40 benchmark.

### F.2  Number of Annealing Stages $N$

The number of annealing stages $N$ determines how gradually the model transitions from the initial annealed distribution toward the target distribution.

In practice:

- more complex or highly multimodal targets generally require larger values of $N$,

- whereas simpler distributions can often be modeled with fewer annealing stages.

Using a sufficiently large $N$ helps maintain overlap between successive intermediate distributions, thereby improving stability of importance-weight estimation and reducing optimization difficulty. Additionally, as shown in Table 5 in Appendix C for the $8 \times 8$ scalar $\phi^4$ distribution, increasing the number of annealing stages $N$ consistently improves performance across all evaluation metrics, indicating that smoother transitions between successive intermediate distributions enhance both distributional fidelity and sampling efficiency.

### F.3   Replay Buffer Size

The replay buffer stores samples generated from previously learned intermediate distributions and is used for importance-weighted supervision during later stages of optimization.

Larger buffer sizes generally improve the stability of importance-weight estimation and enhance mode preservation. However, larger buffers also increase memory requirements.

When the approximate number of modes in the target distribution is known analytically, the replay-buffer size should ideally be chosen large enough to ensure adequate sample representation across all modes. Insufficient buffer coverage may bias the importance-weighted supervision toward dominant modes, reduce stability during later annealing stages, and lead to insufficient mode coverage.

### F.4   Annealing Schedule for $(\lambda_1, \lambda_2)$

The optimization objective in FUND combines the replay-buffer supervision term and the reverse-KL regularization term:

$$\mathcal{L}_{\text{final}}(\theta) = \lambda_1 \mathcal{L}(\theta) + \lambda_2 \mathcal{L}_{\text{RKL}}(\theta).$$

Based on our empirical observations, gradual annealing of the loss weights $(\lambda_1, \lambda_2)$ produces more stable optimization behavior compared to abrupt transitions.

In practice:

- larger initial values of $\lambda_1$ help preserve previously learned mode structure during early optimization,

- while progressively increasing $\lambda_2$ improves local refinement and density correction at later stages.

To systematically study this behavior, we conducted an additional sensitivity analysis on the MW-8 benchmark using multiple annealing schedules. The corresponding observations and quantitative comparisons are presented in Appendix D.

Overall, we observed that smooth transitions between the two objectives improve optimization stability and mode coverage, whereas abrupt changes can degrade performance.

## G   Computational Cost and Runtime Analysis

All experiments were conducted on a system equipped with four NVIDIA GeForce RTX 3080 GPUs, each with 10GB memory. Individual experiments were performed using a single GPU card. We report both:

- total training time required to learn the model, and

- inference/sample-generation time required to generate 10000 samples after training.

We include timing comparisons for both MW-16 and the high-dimensional $\phi^4$ ($12 \times 12$) benchmark. The runtime comparisons are summarized in Table 9.

We did not include FKL in the discussion since it requires access to true samples from the target distribution during training, whereas the remaining methods learn directly from the unnormalized target density without requiring target samples.

The results highlight several important computational trade-offs across methods.

- MCMC methods require computationally expensive sequential sampling during inference, and the computational cost increases significantly with the number of generated samples.

- IDEM has comparatively lower training cost; however, sample generation is substantially slower because inference requires numerically simulating the reverse-time stochastic differential equation (SDE) using the learned diffusion sampler.

Table 9: Training and inference runtime comparison across methods for $\phi^4$ $(12 \times 12)$ and MW-16.

| Method | $\phi^4$ $(12 \times 12)$ | | MW-16 | |
|---|---|---|---|---|
| | Training Time | Inference Time | Training Time | Inference Time |
| MCMC (HMC) | N/A | 510 sec | – | – |
| RKL | 15.16 hrs | 2.2 sec | 0.24 hrs | 0.56 sec |
| FAB | 51.20 hrs | 2.2 sec | 2.5 hrs | 0.56 sec |
| IDEM | 5.25 hrs | 44.0 sec | 4.3 hrs | 34.0 sec |
| FUND(FKL) | 35.69 hrs | 2.2 sec | 2.3 hrs | 0.56 sec |
| FUND(SCORE) | 123.90 hrs | 2.2 sec | 13.6 hrs | 0.56 sec |

- Methods such as FAB and FUND incur higher training cost. In FUND, the additional overhead arises from progressive annealing and repeated optimization across successive intermediate distributions. In FAB, the overhead additionally arises from repeated Annealed Importance Sampling (AIS) steps used for generating training samples.

- Once trained, flow-based approaches enable extremely fast parallel sample generation through a single forward pass of the learned flow model.

- Although RKL requires comparatively moderate training time, it suffers from severe mode-collapse behavior, as reflected in the quantitative results reported in Tables 1 and 2. Consequently, lower training cost alone does not necessarily translate into better generative performance or mode coverage.

- Compared to FAB, FUND(FKL) achieves lower training cost while maintaining similarly fast inference-time generation. In contrast, the score-based variant of FUND incurs higher training overhead due to the additional score-computation operations required for score-loss optimization, including both target-score and model-score evaluations.

- Although FUND introduces additional sequential optimization stages, this overhead is justified by the corresponding improvements in sampling quality and mode coverage. The progressive learning of successive annealed distributions allows the model to incrementally adapt from simpler intermediate densities toward the final target distribution, rather than attempting to learn a highly multimodal target in a single optimization stage. This sequential distributional learning helps preserve modal structure across annealing stages, improves support coverage, and reduces mode-collapse behavior, particularly for challenging multimodal targets. As reflected in Tables 1 and 2, FUND consistently achieves stronger NLL and ESS performance compared to competing methods on several benchmarks.

- Furthermore, once training is complete, FUND enables efficient parallel sample generation through a single forward pass of the flow model, resulting in substantially faster inference compared to MCMC-based approaches, whose sequential sampling cost scales directly with the number of generated samples.

---

**Algorithm 1** Training of FUND

---

1: Initialize flow $q(\mathbf{x}; \theta)$ parameterized by $f$
2: Discretize $t$ into $N$ steps: $\{t_1, t_2, \ldots, t_N\}$
3: Initialize buffer $B \leftarrow \emptyset$
4: **for** $i = 1$ to $N$ **do**
5:     **if** $i = 1$ **then**                                                              ▷ Initial time step
6:         Set $\lambda_1 \leftarrow 0$, $\lambda_2 \leftarrow 1$
7:         **while** $-\mathbb{E}_{q_{t_1}} \log p_{t_1}(x)$ not converged **do**
8:             Sample minibatch $\{z^{(1:m)}\} \sim q_z(z)$
9:             Compute $\mathcal{L}_{\text{RKL}}(\theta)$
10:            Compute:
$$\mathcal{L}_{\text{final}}(\theta) = \lambda_2 \mathcal{L}_{\text{RKL}}(\theta)$$

11:            Update parameters:
$$\theta \leftarrow \theta - \eta \nabla_\theta \mathcal{L}_{\text{final}}(\theta)$$

12:        **end while**
13:        Sample latent variables $z \sim q_z(z)$
14:        Compute $x = f_{t_1}(z; \theta)$
15:        Store $(x, \log q_{t_1}(x))$ in buffer $B$
16:    **else**                                                                         ▷ Subsequent time steps
17:        Compute importance weights for $x_j \in B$:

$$w(x_j) = \frac{p_{t_i}(x_j)}{q_{t_{i-1}}(x_j)}$$

18:        Normalize weights:
$$w(x_j) \leftarrow \frac{w(x_j)}{\sum_{k=1}^{|B|} w(x_k)}$$

19:        Initialize hyperparameters:
$$\lambda_1^{(1)} \leftarrow 1.0, \qquad \lambda_2^{(1)} \leftarrow 0.1$$

20:        **for** $j = 1$ to $K$ **do**                                                 ▷ Annealing stages
21:            **while** $-\mathbb{E}_{q_{t_i}} \log p_{t_i}(x)$ not converged **do**
22:                Sample minibatch $\{(x_b, w_b)\}_{b=1}^{m_B} \sim B$
23:                Sample minibatch $\{z^{(1:m)}\} \sim q_z(z)$
24:                Compute $\mathcal{L}(\theta)$ using minibatch buffer samples $\{(x_b, w_b)\}_{b=1}^{m_B}$
25:                Compute $\mathcal{L}_{\text{RKL}}(\theta)$ using latent minibatch $\{z^{(1:m)}\}$
26:                Compute:
$$\mathcal{L}_{\text{final}}(\theta) = \lambda_1^{(j)} \mathcal{L}(\theta) + \lambda_2^{(j)} \mathcal{L}_{\text{RKL}}(\theta)$$

27:                Update parameters using minibatch gradient:

$$\theta \leftarrow \theta - \eta \nabla_\theta \mathcal{L}_{\text{final}}(\theta)$$

28:            **end while**
29:            Anneal hyperparameters: decrease $\lambda_1^{(j)}$, increase $\lambda_2^{(j)}$
30:        **end for**
31:        Sample latent variables $z \sim q_z(z)$
32:        Compute:
$$x = f_{t_i}(z; \theta)$$

33:        Update buffer $B$ with $(x, \log q_{t_i}(x))$
34:    **end if**
35: **end for**

---

