# OpenReview forum: "FUND: Density Flow for Sampling Unnormalised Distributions"
_TMLR — Decision pending for TMLR_

### Review · Reviewer_yDuR · 2026-03-11

**Summary Of Contributions:**

This paper proposes FUND, a training framework for normalizing flows that targets Boltzmann distributions without requiring any true samples.

The key conceptual departure from prior work is the shift from modeling trajectories of individual samples (as in standard flow matching) to modeling the trajectory of the distribution itself.
The target distribution is gradually shaped as $p_t(x) \propto p(x)^t$, and importance-weighted samples from each intermediate distribution $q_{t_i}$ are reused to train the next stage.
This enables stable sequential learning without access to ground-truth samples.

Compared to FAB, which is bottlenecked by the cost of annealed importance sampling, and iDEM,
which lacks direct likelihood evaluation and requires a two-stage pipeline, FUND uses a single-block normalizing flow that supports exact log-likelihood computation at inference time.

Key contributions:
- A new learning paradigm based on distribution-trajectory matching rather than sample-trajectory matching
- A composite loss combining forward KL and reverse KL with an annealing schedule, balancing mode coverage and mode sharpening

Key strengths:
- ESS remains high even at large lattice sizes ($12 \times 12$), suggesting robustness to increasing dimensionality
- Single-block implementation is memory-efficient, and reproducibility is confirmed across three random seeds
- Figures 3 and 4 provide clear visual evidence of the learning progression

Key weaknesses:
- The hyperparameter tuning process for baseline methods is not described, leaving the fairness of comparisons unclear
- No wall-clock timing comparison is provided, making it difficult to assess whether the sequential overhead is justified
- The claim that the single-block model avoids catastrophic forgetting lacks theoretical justification

**Audience:**

Yes

**Audience Explanation:**

Sampling from Boltzmann distributions is a core challenge in lattice field theory, computational chemistry, and Bayesian inference, all of which have active representation in the TMLR community.
The setting where true samples are unavailable is practically important, since generating them via MCMC or MD is often the computational bottleneck. The concept of distribution-trajectory matching is a genuinely new angle on flow-based samplers, and the application to $\phi^4$ theory provides a concrete and well-established testbed.
The findings are likely to be of interest to researchers working on both the methodological and application sides.

**Claims And Evidence:**

No

**Claims Explanation:**

The numerical results in Tables 1 and 2 are broadly consistent with the paper's main claims, but several aspects of the experimental setup are underspecified, making it difficult to fully assess the validity of the reported comparisons.

### 1. Fairness of baseline comparisons

The most pressing concern is the asymmetry in hyperparameter tuning.
For FUND, the paper describes grid search over $t_1$, ablations over $N$ (Table 5), buffer size selection, and the $\lambda_1$/$\lambda_2$ annealing schedule.
For FAB and iDEM, no equivalent information is given.
If these baselines were run at or near default settings, a nontrivial portion of the reported performance gap may reflect tuning asymmetry rather than algorithmic advantage.
At a minimum, the paper should state whether the recommended configurations from the original publications were used, or whether equivalent computational budgets were allocated for tuning each method.

### 2. Lack of wall-clock timing

The paper acknowledges that FUND's sequential learning scales linearly with the number of steps $N$, but provides no runtime comparison against the baselines. Without this, claims about scalability and efficiency are one-sided.
For the largest benchmark ($12 \times 12$ $\phi^4$, $N=9$), it is unclear how FUND's training time compares to FAB or iDEM running under similar conditions.

### 3. Missing error bars in the main tables

Tables 1 and 2 report single values with no uncertainty estimates.
The seed-level results in Appendix D show non-negligible variance: for instance, ESS of FUND(FKL) on the $8 \times 8$ $\phi^4$ lattice ranges from 0.08 to 0.19 across seeds. Given this variability, some of the margins in the main tables may not be statistically meaningful.
Standard deviations should be included directly in Tables 1 and 2.

### 4. Catastrophic forgetting claim lacks theoretical backing

Section 2.3 states that $L(\theta)$ discourages catastrophic forgetting, but this is not supported by any formal argument.
The current presentation treats this as an empirically observed fact, which is insufficient given how central this property is to the method's design.

**Requested Changes:**

[R1] Describe the hyperparameter tuning procedure for baseline methods

For FAB and iDEM, specify whether the configurations used correspond to the recommended settings from the original papers, or whether a search was conducted with a comparable budget to what was used for FUND. Without this, the fairness of the comparisons in Tables 1 and 2 cannot be assessed.

[R2] Add error bars to Tables 1 and 2

Report mean and standard deviation (or standard error) across multiple seeds for at least NLL and ESS in the main tables. The current placement of seed-level results in Appendix D is insufficient.
Readers need to be able to judge statistical significance of the reported differences directly from the main results.

[R3] Provide wall-clock timing comparisons

Report training and inference times for all compared methods under the same hardware setup (including GPU type and count) for at least MW-16 and $\phi^4$ $12 \times 12$.
The paper should show that FUND's sequential overhead is justified by the performance gains, rather than leaving this as an implicit assumption.

[R4] Strengthen the justification for avoiding catastrophic forgetting

Section 2.3 should go beyond the current empirical observation.
Even a brief analytical argument showing how the importance-weighted loss preserves information from earlier stages would meaningfully support this claim.

[R5] Provide a sensitivity analysis for $t_1$ selection

The paper acknowledges that an overly large $t_1$ causes mode collapse, but does not characterize how sensitive the method is to this choice.
A systematic experiment varying $t_1$ on MOG-40 (e.g., tracking how many modes are recovered as a function of $t_1$) would help readers understand the practical limits of the method.

[R6] Add practical guidance for hyperparameter selection

Given the number of hyperparameters involved ($N$, $t_1$, buffer size, $\lambda$ schedule), an appendix section offering rough guidelines for new target distributions would improve the usability of the method.
For example, relating $t_1$ to a rough characterization of multimodality would be a reasonable starting point.

[C7] Include a direct computational cost comparison across methods

Section 4 should contain a brief discussion comparing the computational profiles of FUND, FAB, and iDEM in qualitative or quantitative terms.
Each method carries its own overhead (sequential training, AIS, SDE solving), and helping readers understand the cost-accuracy tradeoff would make the contribution clearer.

---

> ### Author Response · Authors · 2026-05-21
>
> [R1] Describe the hyperparameter tuning procedure for baseline methods
>
> For FAB and iDEM, specify whether the configurations used correspond to the recommended settings from the original papers, or whether a search was conducted with a comparable budget to what was used for FUND. Without this, the fairness of the comparisons in Tables 1 and 2 cannot be assessed.
>
> > We thank the reviewer for raising this important point regarding experimental fairness.
> >
> > For both FAB and iDEM, we primarily followed the architectures, optimizer settings, and training procedures recommended in the respective original papers and official repositories. In addition, we performed limited hyperparameter tuning to ensure stable convergence and competitive performance on our benchmark distributions.
> >
> >Since both FAB and FUND are flow-based methods, we kept the normalizing-flow (NF) architectures identical wherever possible to ensure a fair comparison. In particular, the NF configurations for FUND and FAB were matched for the scalar $\phi^4$ experiments. Complete architectural details are provided in Table 4 of Appendix B.
> >
> >For MOG-40, MW-8 and MW-16, we used the recommended settings reported in the original FAB paper and official implementation repository.
> >
> >For FAB, hyperparameter tuning primarily involved the number of intermediate annealing distributions and the number of MCMC transition steps. In the $\phi^4$ experiments, we used HMC as the transition operator with 4 intermediate distributions and 5 MCMC transition steps. The learning rate was fixed at $1\times10^{-4}$.
> >
> >Similarly, for IDEM, we largely followed the official implementation and only performed limited tuning for the $\phi^4$ experiments, including learning rate, batch size, and model depth/width within reasonable ranges. We used an MLP architecture consisting of 4 hidden layers with 512 neurons per layer and sinusoidal embeddings of dimension 512. The learning rate was fixed at $1\times10^{-4}$. The remaining hyperparameter settings were adopted from the original repository.
> >
> >For the remaining baselines, namely FKL and RKL, we used the same RealNVP architecture as employed in FUND to ensure a fair comparison across flow-based models.
> >
> >We did not perform exhaustive large-scale hyperparameter searches for any method. All methods were trained under comparable computational budgets and hardware settings to ensure a fair empirical comparison. The rest of the hyperparameter details would be subsequently shared on the FUND github repo asap.
>
>
> [R2] Add error bars to Tables 1 and 2
>
> Report mean and standard deviation (or standard error) across multiple seeds for at least NLL and ESS in the main tables. The current placement of seed-level results in Appendix D is insufficient. Readers need to be able to judge statistical significance of the reported differences directly from the main results.
>
> > We agree that reporting only single-run results in the main tables makes it difficult to assess the statistical significance and robustness of the reported improvements.
> >
> > Accordingly, we have revised Tables 1 and 2 to report the mean and standard deviation across three random seeds for the primary evaluation metrics, including NLL and ESS. The seed-level experimental results previously provided in Appendix D are now summarized directly in the main tables for improved clarity and comparability.

---

> > ### Author Response · Authors · 2026-05-21
> >
> > [R3] Provide wall-clock timing comparisons
> >
> > Report training and inference times for all compared methods under the same hardware setup (including GPU type and count) for at least MW-16 and $\phi^4$ (12x12). The paper should show that FUND's sequential overhead is justified by the performance gains, rather than leaving this as an implicit assumption.
> >
> > > We thank the reviewer for this important suggestion. In the revised manuscript, we have included explicit wall-clock training and inference time comparisons for all methods evaluated under the same hardware configuration.
> > >
> > > All experiments were conducted on a system equipped with four NVIDIA GeForce RTX 3080 GPUs, each with 10GB memory. Individual experiments were performed using a single GPU card. We report both:
> > > - total training time required to learn the model, and
> > > - inference/sample-generation time required to generate 10000 of samples after training.
> > >
> > > We include timing comparisons for both MW-16 and the high-dimensional $\phi^4$ $(12\times12)$ benchmark. The results for $\phi^4$ $(12\times12)$ and MW-16 are summarized below:
> > >
> > > For **$\phi^4$ (12\times12)**
> > > | Method | Training Time | Inference / Generation Time |
> > > |---|---:|---:|
> > > | MCMC (HMC Algorithm) | 0 sec | 510 sec |
> > > | RKL | 15.16 hrs | 2.2 sec |
> > > | FAB | 51.20 hrs | 2.2 sec |
> > > | iDEM | 5.25 hrs | 44.0 sec |
> > > | FUND (FKL) | 35.69 hrs | 2.2 sec |
> > > | FUND (Score) | 123.90 hrs | 2.2 sec |
> > >
> > > For **MW-16 distribution**
> > | Method | Training Time | Inference / Generation Time |
> > |---|---:|---:|
> > | RKL | 0.24 hrs | 0.56 sec |
> > | FAB | 2.5 hrs | 0.56 sec |
> > | iDEM | 4.3 hrs | 34.0 sec |
> > | FUND (FKL) | 2.3 hrs | 0.56 sec |
> > | FUND (Score) | 13.6 hrs | 0.56 sec |
> > > We did not include FKL in the discussion since it requires access to true samples from the target distribution during training, whereas the remaining methods learn directly from the unnormalized target density without requiring target samples.
> > >
> > > The results highlight several important trade-offs:
> > >
> > > - MCMC methods require computationally expensive sequential sampling during inference, and the cost increases significantly with the number of generated samples.
> > >
> > > - iDEM has comparatively lower training cost; however, sample generation is slower because inference requires simulating the reverse-time stochastic differential equation (SDE) using the learned diffusion sampler.
> >
> > > - Methods such as FAB and FUND incur higher training cost. In FUND, the additional overhead arises from progressive annealing and repeated optimization across successive intermediate distributions. In FAB, this overhead additionally arises from repeated Annealed Importance Sampling (AIS) steps used for generating training samples.
> > > - However once trained, flow-based approaches enable extremely fast parallel sample generation through a single forward pass,
> >
> > > - We also note that while RKL requires comparatively moderate training time, it suffers from severe mode-collapse behavior, as reflected in the quantitative results reported in Tables 1 and 2. Consequently, lower training cost alone does not necessarily translate into better generative performance or mode coverage.
> > >
> > > - Compared to FAB, FUND (FKL) achieves lower training cost while maintaining similarly fast inference-time generation. In contrast, Score variant of FUND incur higher training overhead due to the additional score-computation (both target as well as model score via gradient operation) in score loss optimization.
> > >
> > >- Although FUND introduces additional sequential optimization stages, this overhead is justified by the corresponding improvements in sampling quality and mode coverage. The progressive learning of successive annealed distributions allows the model to incrementally adapt from simpler intermediate densities toward the final target distribution, rather than attempting to learn a highly multimodal target in a single step. This sequential distributional learning helps preserve the modal structure across annealing stages, improves support coverage, and reduces mode-collapse behavior, particularly for challenging multimodal targets. As reflected in Tables 1 and 2, FUND consistently achieves stronger NLL/ESS performance compared to competing methods on several benchmarks. Furthermore, once training is complete, FUND enables efficient parallel sample generation through a single forward pass of the flow model, resulting in substantially faster inference than MCMC-based approaches, whose sequential sampling cost scales directly with the number of generated samples.
> > > We have included these runtime comparisons and hardware details in the revised manuscript to provide a clearer picture of the computational trade-offs between methods.

---

> > > ### Author Response · Authors · 2026-05-21
> > >
> > > [R4] Strengthen the justification for avoiding catastrophic forgetting
> > >
> > > Section 2.3 should go beyond the current empirical observation. Even a brief analytical argument showing how the importance-weighted loss preserves information from earlier stages would meaningfully support this claim.
> > >
> > > > We thank the reviewer for this observation. We agree that the discussion in Section 2.3 should provide stronger justification beyond empirical observations alone.
> > > >
> > > > Our motivation is based on the well-established role of replay buffers in continual learning, where previously observed samples are reused during subsequent optimization steps to mitigate catastrophic forgetting and reduce distributional drift [1,2].
> > > >
> > > > In FUND, the persistent sample buffer continuously reuses samples from earlier intermediate distributions while optimizing later annealing stages. Consequently, the model is not trained solely on the current target distribution at each stage, but is repeatedly exposed to samples representing previously learned modes.
> > > >
> > > > More specifically, the optimization objective
> > > >
> > > > $ \mathcal{L_{final}}(\theta) = \lambda_1 \mathcal{L}(\theta) + \lambda_2 \mathcal{L_{RKL}}(\theta)$
> > > >
> > > > combines the reverse KL term with either an FKL- or score-based supervision objective. The reverse KL term uses samples generated from the current model, while the supervision term operates on replay-buffer samples obtained from previously learned intermediate distributions.
> > > >
> > > > Since both FKL- and score-based objectives encourage probability mass coverage across multiple modes [3,4], the replay-buffer supervision continually reinforces previously discovered regions of the target distribution during later optimization stages. This reduces the tendency of the model to drift toward newly emphasized modes and empirically mitigates catastrophic forgetting.
> > > >
> > > > Accordingly, we have revised Section 2.3  to provide a clearer explanation of this mechanism with relevant supporting citations.
> > >
> > > >[1] Timothée Lesort. *Continual learning: Tackling catastrophic forgetting in deep neural networks with replay processes*. arXiv preprint arXiv:2007.00487, 2020.
> > > >
> > > >[2] David Rolnick, Arun Ahuja, Jonathan Schwarz, Timothy Lillicrap, and Gregory Wayne. *Experience replay for continual learning*. **Advances in Neural Information Processing Systems**, 32, 2019.
> > > >
> > > >[3] Vikas Kanaujia and Vipul Arora. *Scorenf: Score-based normalizing flows for sampling unnormalized distributions*. In **ICASSP 2026 – 2026 IEEE International Conference on Acoustics, Speech and Signal Processing (ICASSP)**, pages 4336–4340. IEEE, 2026.
> > > >
> > > >[4] Vikas Kanaujia, Mathias S. Scheurer, and Vipul Arora. *Advnf: Reducing mode collapse in conditional normalising flows using adversarial learning*. **SciPost Physics**, 16(5):132, 2024.
> > >
> > > [R5] Provide a sensitivity analysis for $t_1$ selection
> > > The paper acknowledges that an overly large $t_1$ causes mode collapse, but does not characterize how sensitive the method is to this choice. A systematic experiment varying  on MOG-40 (e.g., tracking how many modes are recovered as a function of $t_1$) would help readers understand the practical limits of the method.
> > >
> > > > We thank the reviewer for this valuable suggestion.
> > > >
> > > > The choice of the initial annealing parameter $t_1$ plays an important role in the stability of the method and the preservation of mode coverage during the early stages of training.
> > > >
> > > > As discussed in the manuscript, choosing a sufficiently small $t_1$ smooths the target landscape and connects otherwise isolated modes, making the initial optimization problem easier. However, when $t_1$ is chosen too large, the initial annealed distribution remains highly multimodal, which can increase the mode-seeking tendency of reverse KL optimization and lead to mode collapse.
> > > >
> > > > To better characterize this behavior, we conducted additional sensitivity experiments on the MOG-40 benchmark by varying $t_1$ and measuring the number of recovered modes together with the corresponding NLL  value.
> > > >|              | NLL($\downarrow$) | No. of Modes Recovered |
> > > >|--------------|-------------------|------------------------|
> > > >| $t_1 = 0.10$ | 1327.82           | 12                     |
> > > >| $t_1 = 0.01$ | 7.07              | 40                     |
> > > >
> > > > Our preliminary observations indicate a clear trend:
> > > >
> > > > - larger values of $t_1$ lead to mode collapse,
> > > > - while sufficiently small values of $t_1$ significantly improve stability and preserve the modal structure of the target distribution.
> > > >
> > > > For example, we observed noticeable mode collapse at $t_1 = 0.1$, whereas using $t_1 = 0.01$ preserved all major modes reliably.
> > > >
> > > > we have included this sensitivity analysis in the revised manuscript (appendix) showing how performance and mode recovery vary as a function of $t_1$ on the MOG-40 dataset including figure as well.

---

> > > > ### Author Response · Authors · 2026-05-21
> > > >
> > > > [R6] Add practical guidance for hyperparameter selection
> > > >
> > > > > We thank the reviewer for this helpful suggestion.
> > > > >
> > > > > We agree that practical guidance for selecting hyperparameters would improve the usability and reproducibility of the method, particularly for new target distributions with unknown structure.
> > > > >
> > > > > Based on our empirical observations, the most important hyperparameter is the initial annealing parameter $t_1$, since it controls the smoothness of the initial intermediate distribution. In general:
> > > > >
> > > > > - highly multimodal distributions with sharp energy barriers benefit from smaller values of $t_1$,
> > > > > - whereas simpler or weakly multimodal targets can tolerate relatively larger values.
> > > > >
> > > > > Intuitively, $t_1$ should be chosen small enough so that the initial annealed distribution has sufficient overlap across modes and can be modeled without severe mode collapse.
> > > > >
> > > > > Similarly:
> > > > >
> > > > > - the number of annealing stages $N$ should increase with target complexity to maintain sufficient overlap between successive distributions. In Table 5, we report results for the scalar $\phi^{4}$ model distribution on an $8 \times 8$ lattice, illustrating the effect of varying the number of intermediate distributions $N$ on model performance.
> > > > > - larger buffer sizes improve stability of importance-weight estimation but increase memory cost. Furthermore, when the approximate number of modes in the target distribution is known analytically, the replay-buffer size should be chosen large enough to ensure adequate representation of samples across all modes. Insufficient buffer coverage may bias the importance-weighted supervision toward dominant modes and reduce stability during later annealing stages and result in insufficient mode coverage.
> > > > > - gradual annealing of the loss weights $(\lambda_1,\lambda_2)$ was observed to produce more stable optimization behavior compared to abrupt transitions. To study this effect systematically, we conducted an additional sensitivity analysis on the MW-8 distribution using the FUND(FKL) variant under four different annealing schedules. The corresponding observations and conclusions are discussed in our response to Reviewer L2yo.(Refer Appendix D \& Table 7 in the revised paper)
> > > > >
> > > > > Based on the reviewer’s suggestion, we have added an appendix section summarizing practical guidelines and heuristics for selecting:
> > > > >
> > > > > - $t_1$,
> > > > > - the number of annealing stages $N$,
> > > > > - replay-buffer size,
> > > > > - and the $(\lambda_1,\lambda_2)$ annealing schedule.
> > > > >
> > > >
> > > >
> > > > [C7] Include a direct computational cost comparison across methods
> > > >
> > > > Section 4 should contain a brief discussion comparing the computational profiles of FUND, FAB, and iDEM in qualitative or quantitative terms. Each method carries its own overhead (sequential training, AIS, SDE solving), and helping readers understand the cost-accuracy tradeoff would make the contribution clearer.
> > > >
> > > >
> > > > > We thank the reviewer for this valuable suggestion. In the revised manuscript, we have included an additional discussion comparing the computational characteristics of FUND, FAB, and iDEM, since each method introduces a different form of computational overhead.
> > > > >
> > > > > FUND introduces additional sequential optimization stages due to progressive annealing across intermediate distributions. However, this staged optimization significantly improves stability and mode coverage by gradually transporting the model from simpler intermediate densities toward the final multimodal target distribution. As observed in Tables 1 and 2, this sequential distributional learning improves NLL and ESS performance while mitigating mode-collapse behavior. Once training is complete, FUND enables efficient parallel sample generation through a single forward pass of the trained flow model.
> > > > >
> > > > > FAB incurs additional computational overhead from Annealed Importance Sampling (AIS), which is repeatedly used to generate samples required during optimization. This substantially increases training cost, particularly in higher-dimensional settings.
> > > >
> > > > > In contrast, iDEM has comparatively lower training cost but slower inference. Sample generation requires numerically simulating the reverse-time stochastic differential equation (SDE) using the learned diffusion sampler, making inference substantially more expensive than direct flow-based sampling approaches.
> > > > >
> > > > > MCMC-based methods exhibit the opposite trade-off: they require little or no training cost but incur high inference cost because samples must be generated sequentially through Markov chain transitions, with computational cost scaling directly with the number of generated samples.
> > > > >
> > > > > Overall, the experiments suggest that FUND achieves a favorable cost-accuracy trade-off: although training is more expensive than single-stage flow optimization, the sequential annealing strategy produces substantially improved sample quality and mode coverage while still retaining fast parallel inference after training.

---

> > > > > ### Comment · Reviewer_yDuR · 2026-06-01
> > > > >
> > > > > I thank the authors for the detailed response and the revised manuscript. The revision addresses most of my main concerns.
> > > > >
> > > > > The added description of the baseline setup makes the comparisons clearer. I appreciate that the authors matched the flow architecture between FUND and FAB where possible and followed the official configurations for FAB and iDEM, with only limited additional tuning. I still recommend that the final manuscript include all important baseline hyperparameters in the appendix, rather than deferring some details to the GitHub repository.
> > > > >
> > > > > The added mean and standard deviation in Tables 1 and 2 are useful and address my concern about uncertainty in the main results.
> > > > >
> > > > > The runtime table is also helpful. It shows that the strength of FUND is not that it is uniformly faster than existing methods. FUND is better viewed as offering a useful tradeoff between sample quality, mode coverage, exact likelihood evaluation, and fast inference. In particular, FUND(FKL) is competitive with FAB in training time and keeps the fast inference of flow-based models. FUND(SCORE) gives the strongest sampling results, but its training cost is much higher, especially on the phi^4(12x12) benchmark. I therefore recommend tempering broad efficiency claims and clearly separating training time, inference time, and sample quality. A precise statement would be that FUND(FKL) has a favorable cost-quality tradeoff, while FUND(SCORE) is a higher-quality but more expensive variant.
> > > > >
> > > > > The additional discussion of the replay buffer is reasonable and clarifies why FUND may reduce catastrophic forgetting. I would still avoid language that suggests a formal guarantee. "Mitigates catastrophic forgetting" would be more appropriate than "prevents catastrophic forgetting."
> > > > >
> > > > > The added sensitivity analysis for t1 and the practical guidelines are useful. The current t1 experiment is informative, but it only compares two values. I would describe it as an illustrative sensitivity check rather than a systematic study.
> > > > >
> > > > > Overall, I am satisfied with the revision. My remaining concerns are minor and can be addressed by making the final manuscript self-contained, tempering the efficiency and forgetting-related claims, and fixing small presentation issues such as the duplicated phi^4(10x10) row in Table 6 and the "0 sec" training time for HMC, which may be clearer as "N/A."

---

> ### Author Response · Authors · 2026-06-14
>
> 1. The added description of the baseline setup makes the comparisons clearer. I appreciate that the authors matched the flow architecture between FUND and FAB where possible and followed the official configurations for FAB and iDEM, with only limited additional tuning. I still recommend that the final manuscript include all important baseline hyperparameters in the appendix, rather than deferring some details to the GitHub repository.
>
> > We thank the reviewer for the suggestion. We have revised the manuscript and added the important baseline hyperparameters in Appendix B (Model Details).
>
> 2. The added mean and standard deviation in Tables 1 and 2 are useful and address my concern about uncertainty in the main results. The runtime table is also helpful. It shows that the strength of FUND is not that it is uniformly faster than existing methods. FUND is better viewed as offering a useful tradeoff between sample quality, mode coverage, exact likelihood evaluation, and fast inference. In particular, FUND(FKL) is competitive with FAB in training time and keeps the fast inference of flow-based models. FUND(SCORE) gives the strongest sampling results, but its training cost is much higher, especially on the phi^4(12x12) benchmark. I therefore recommend tempering broad efficiency claims and clearly separating training time, inference time, and sample quality. A precise statement would be that FUND(FKL) has a favorable cost-quality tradeoff, while FUND(SCORE) is a higher-quality but more expensive variant.
>
> >We thank the reviewer for the valuable feedback. In response, we have revised the manuscript to moderate broad efficiency claims and to more clearly highlight the distinct cost–quality tradeoffs associated with the two variants of the proposed framework in the abstract, results, and conclusion sections.
>
> 3. The additional discussion of the replay buffer is reasonable and clarifies why FUND may reduce catastrophic forgetting. I would still avoid language that suggests a formal guarantee. "Mitigates catastrophic forgetting" would be more appropriate than "prevents catastrophic forgetting."
>
> >We thank the reviewer for the suggestion. We agree that the original wording was too strong. Accordingly, we have revised the manuscript to replace “prevents catastrophic forgetting” with “mitigates catastrophic forgetting” wherever applicable.
>
> 4. The added sensitivity analysis for t1 and the practical guidelines are useful. The current t1 experiment is informative, but it only compares two values. I would describe it as an illustrative sensitivity check rather than a systematic study.
>
> > We thank the reviewer for the suggestion. To provide a more comprehensive sensitivity analysis, we have expanded the experiment to include five values of the initial annealing parameter, $t_1 \in {0.01, 0.025, 0.05, 0.075, 0.1}$. The results show a clear trend: increasing $t_1$ leads to degradation in NLL and reduced mode recovery, indicating increased mode collapse. We have added these results and the corresponding discussion to the revised manuscript.
> >| $t_1$ | NLL($\downarrow$) | Number of Modes Recovered |
> >|-------|-------------------|---------------------------|
> >| 0.010 | 7.07              | 40                        |
> >| 0.025 | 20.03             | 33                        |
> >| 0.050 | 53.20             | 20                        |
> >| 0.075 | 990.99            | 15                        |
> >| 0.100 | 1327.82           | 12                        |
>
> 5. Overall, I am satisfied with the revision. My remaining concerns are minor and can be addressed by making the final manuscript self-contained, tempering the efficiency and forgetting-related claims, and fixing small presentation issues such as the duplicated phi^4(10x10) row in Table 6 and the "0 sec" training time for HMC, which may be clearer as "N/A."
>
> > We thank the reviewer for the helpful suggestions and have addressed all the points raised in the revised manuscript.

---

### Review · Reviewer_L2yo · 2026-04-22

**Summary Of Contributions:**

The work presents a method for learning to sample from an unnormalised probability distribution using evaluations, rather than samples from the true distribution, which would require methods like MCMC or annealed Langevin dynamics that are expensive and suffer from problems like missing modes and sample correlations. An example application is the common problem of sampling from a Boltzmann distribution for which we know the energy function and temperature (a.k.a. energy-based modelling). This occurs in settings such as computational physical sciences and Bayesian posterior estimation.

The method uses a recursively called normalising flow model (although one can implement multiple models in a chain) to iteratively transform a single-mode base distribution into the target. The transformations are trained to match a supervised temperature annealing schedule $p_t(x) \propto p(x)^t$ using discrete timesteps on the domain $t\in(0,1]$, without requiring access to normalising constants. To avoid an intractable expectation over $p_t$, each step is trained using importance sampling wrt an earlier timestep. The annealing schedule is important for constraining the variance of importance weights, which is needed for numerical stability.

Key strengths:
- Important problem setting
- Methodology appears sound
- Positive proof-of-principle results in experimental settings

Key weaknesses:
- Title promises strong connection with flow-matching but actual method implements discrete time mappings, not continuous time velocities, which are a hallmark of flow-matching.
- Experimental evidence could be stronger in terms of problem difficulty/realism, uncertainty analysis, and visualisation of method behaviours.

**Additional Comments:**

Please answer the following additional questions and comments, and clarify them in the text where appropriate.

**Framing and positioning**

- Can any other method optimise a non-rKL objective for sampling energy-based distributions without true samples or MCMC/Langevin?
- Can you explain exactly how AIS limits FAB in terms of its computational complexity, and scaling to higher dimensions? How we expect your method to compare on each of these points?

**Methodology**

- We must choose small $t_1$ to achieve connected modes in $p_{t_1}$, but this could make the prior very disperse compared with $p(x)$ - especially when data modes are separated by large barriers. Can this cause problems with the variance of the importance weights, since it has a long way to travel to eventually reach $p(x)$? Other factors that might affect the learnability of $q_{t_1}$ are (i) the fact that (without samples) the model does not know what a sensible initial state is, and may be initially very unrepresentative of $p_{t_1}$ (barely overlapping supports), and (ii) the fact that $q_{t_1}$ is trained with rKL, which can cause it to mode-seek over $p_{t_1}$, reducing the effective support. Do you observe any such effects in your experiments? Do you expect there to be situations in which these are problems?
- Importance sampling becomes inefficient in high-dimensions. How does this affect the scalability of the method to high-dimensions?

**Evidence**

- Abstract: "Our approach not only outperforms traditional MCMC and flow-based methods in efficiency..." What do you mean by improved efficiency in this claim, and what is the supporting evidence? Furthermore, the results do not show any comparisons with MCMC.

**Audience:**

Yes

**Audience Explanation:**

Sampling from energy-based models with intractable normalisation is a classic problem. New methods that avoid MCMC or Langevin dynamics, complicated score-matching, and providing a balance of mode covering properties are welcome.

**Claims And Evidence:**

No

**Claims Explanation:**

I think this criteria is mostly met. I just have a couple of outstanding concerns to address, particularly:

1. As noted above, I don't think it's accurate to name this as a flow-matching method in the title, since it is learning discrete time mappings instead of continuous time velocity field.
2. Whether the evidence is sufficiently strong to support strong claims like "The results demonstrate that our method substantially improves deep learning–based sample generation for unnormalized targets, enabling efficient training without access to true samples— that would otherwise require computationally expensive MCMC or molecular-dynamics (MD) simulations.", "We validate our method on challenging benchmarks", or "establishes a new paradigm for sample-free learning of complex physical distributions." I favour more tempered language claiming competitive performance, and without overclaiming the strength of the three relatively low-dimensional synthetic experimental settings.

**Requested Changes:**

**Framing and positioning**

- [Essential] The narrative places a strong emphasis on a connection with flow matching. However, I understand that flow matching typically means learning a _continuous time average velocity field_ that we numerically integrate to propagate a noise sample onto the target. This work does not learn+integrate any continuous time object (in this case, a distribution instead of sample). Instead, the method learns a series of discrete time mappings along the annealed trajectory $p_t(x)\propto p(x)^t$ for $t\in(0,1]$. Can we justify the "flow matching" name, or is this misleading? If we cannot justify it then I would ask for this naming to be changed.

**Methodology**

- [Essential] The definition of the importance weights (equation 3) and losses (equations 3 and 7) assume access to a normalised $p_t$, which is unavailable. Please state exactly the equations that allow all losses to be calculated using the unnormalised distribution or energy function (the importance weights are self-normalised, and the score is not sensitive to the norm constant).
- Figure 1 caused me quite a bit of confusion and I suggest relocating it [optional] + improving the clarity [essential]. The placement was confusing because it contradicts the adjacent text of section 2, which uses a continuous time subscript and makes no mention of a recursive model. Furthermore, Figure 1 shows a conflicting picture in which the same NF sometimes contains two inputs (both $z$ and $x_t$), which does not match anything in the text, which only describes one-to-one mappings. I still don't understand why you show $z$ as input at every timestep - please can you explain this?
- [Essential] Figure 1 describes a single NF parametrised by parameters $\theta$ that is re-used for every time increment. Please explain why it is valid to re-use the parameters $\theta$ at different timesteps, since these represent different mappings. Are there scenarios in which it would not be? This is especially confusing because your $\delta t$ are not always the same (Table 6), and so $\frac{p_{t_{i+1}}(x)}{p_{t_i}(x)}$ is not independent of $i$. Furthermore, does updating the parameters at later timesteps not destabilise the behaviour learned at earlier timesteps?

**Evidence**

- [Essential] Temper claims like "our method substantially improves deep learning–based sample generation" in favour of language that shows competitive performance in the settings shown.
- The work would be strengthened by uncertainties on the numbers in Tables 1 and 2, describing the variance over training seeds as well as the statistical precision of the test data. This would reveal whether performance differences are statistically significant.
- Where data are 2D or below (c.f. MOG-40 and $\phi^4$), the work would be strengthened by visualisations of samples from each of the models in an appendix, alongside the ground truth distribution. This would strengthen the case for interpretations of mode-missing and over-covering behaviour, beyond the numbers in the Tables.

---

> ### Author Response · Authors · 2026-05-21
>
> **Framing and Positioning**
> > We thank the reviewer for raising this important point regarding the terminology.
> >
> > We agree that the term *flow matching* is commonly associated with methods that learn a **continuous-time velocity field** whose numerical integration transports samples from a base distribution to a target distribution. In contrast, our method does **not** parameterize or integrate a continuous-time vector field. Instead, it learns a sequence of discrete transport maps corresponding to intermediate annealed distributions
> >
> > $$
> > p_t(x) \propto p(x)^t, \qquad t \in (0,1].
> > $$
> >
> > Our use of the term “flow” was motivated by the fact that the method constructs a progressive transport of probability mass through a sequence of invertible mappings across the annealing trajectory. However, we acknowledge that this differs from the standard usage of *flow matching* in the recent literature, where the emphasis is specifically on continuous-time dynamics and velocity-field matching.
> >
> > We therefore agree that the current terminology may create confusion for readers familiar with contemporary continuous-time flow matching frameworks.
> >
> > To avoid ambiguity, we have revised the terminology throughout the paper and avoid referring to the method as “flow matching.” Instead, we will describe the approach more precisely as:
> >
> > - **Density Flow for Sampling Unnormalised Distribution**
> >
> > We will also explicitly clarify in the revised manuscript that our method:
> >
> > - does **not** learn a continuous-time velocity field,
> > - does **not** require ODE/SDE integration, and
> > - instead learns discrete mappings between successive annealed distributions.
> >
> > We thank the reviewer for pointing out this distinction, which will help improve the clarity and positioning of the work.
>
> **Methodology**
> [Essential] The definition of the importance weights (equation 3) and losses (equations 3 and 7) assume access to a normalised
> , which is unavailable. Please state exactly the equations that allow all losses to be calculated using the unnormalised distribution or energy function (the importance weights are self-normalised, and the score is not sensitive to the norm constant).
>
> > In our setting, the target distribution is defined as
> >
> > $ p(x) = \frac{\tilde{p}(x)}{Z} \propto \exp(-E(x)),$
> >
> > where $\tilde{p}(x)$ denotes the unnormalized density and $Z$ is the unknown partition function.
> >
> > The intermediate annealed distributions are therefore:
> >
> > $ p_t(x) = \frac{\tilde{p}(x)^t}{Z_t}$
> >
> > Importantly, all optimization objectives used in FUND can be evaluated without knowledge of the normalization constants.
> >
> > For the importance weights in Eq. (3), we use self-normalized importance sampling:
> >$w(x_j)=\frac{p_{t_i}(x_j)}{q_{t_{i-1}}(x_j)}$
> >
> > followed by normalization:
> >
> > $ \bar{w}(x_j) = \frac{w(x_j)} {\sum_k w(x_k)}$
> >
> > Since the unknown partition constant appears identically in all weights, it cancels during normalization and therefore does not need to be computed.
> >
> > Similarly, the reverse KL objective can be written as:
> >
> > $ \mathcal{L_{RKL}}(\theta) = \mathbb{E}_{q_t} \left[ \log q_t(x) - \log \tilde{p}_t(x) \right] + \log Z_t $
> >
> > The additive term $\log Z_t$ is independent of model parameters $\theta$ and therefore does not affect gradient-based optimization. Consequently, optimization only requires evaluating the unnormalized density (or equivalently the energy function).
> >
> > Likewise, score-based objectives depend only on:
> >
> > $ \nabla_x \log p_t(x) = \nabla_x \log \tilde{p}_t(x),$
> >
> > making them inherently insensitive to the unknown normalization constant.

---

> > ### Author Response · Authors · 2026-05-21
> >
> > ###Methodology
> > 3. [Essential] Figure 1 describes a single NF parametrised by parameters $\theta$ that is re-used for every time increment. Please explain why it is valid to re-use the parameters at different timesteps, since these represent different mappings. Are there scenarios in which it would not be? This is especially confusing because your $\delta t$ are not always the same (Table 6), and so $\frac{p_{t_{i+1}}(x)}{p_{t_i}(x)}$ is not independent of $i$. Furthermore, does updating the parameters at later timesteps not destabilise the behaviour learned at earlier timesteps?
> >
> > > We thank the reviewer for this important observation. We agree that Figure 1 and the accompanying discussion did not sufficiently clarify the rationale behind reusing the same NF parameters across annealing stages.
> > >
> > > In our framework, we considered two possible implementations.
> > >
> > > One possibility is an incremental architecture in which a new NF coupling block is appended whenever $t$ is incremented, while previously learned blocks are frozen. Although this explicitly separates the mappings across timesteps, we found this design to be memory-inefficient and empirically less effective at learning all modes of the target distribution (details are provided in Appendix B.
> > >
> > > Instead, we adopt a single-block architecture that is progressively refined across all annealing stages. Training proceeds sequentially:
> > >
> > > - the model is first optimized as $q_{t_1}$ with $p_{t_1}$ as the target,
> > > - and at subsequent stages, the same model parameters are further optimized to represent $q_t$ with $p_t$ as the target.
> > >
> > > Thus, parameter reuse should be interpreted as a warm-start or continuation strategy rather than assuming that all intermediate mappings are identical.
> > >
> > > Since consecutive annealed distributions
> > >
> > > $$
> > > p_{t_i}(x) \propto p(x)^{t_i}
> > > $$
> > >
> > > and
> > >
> > > $$
> > > p_{t_{i+1}}(x) \propto p(x)^{t_{i+1}}
> > > $$
> > >
> > > are typically close for sufficiently small annealing increments, the mapping learned at timestep $t_i$ provides a useful initialization for learning the next mapping at $t_{i+1}$. This substantially reduces memory usage and improves optimization efficiency compared to training separate models from scratch at every stage.
> > >
> > > We agree with the reviewer that the transport between successive stages is not timestep-invariant, especially when the annealing increments $\delta t$ are non-uniform as in Table 6. In such cases,
> > >
> > > $$
> > > \frac{p_{t_{i+1}}(x)}{p_{t_i}(x)}
> > > $$
> > >
> > > indeed depends on the specific stage index $i$. Our method does not assume identical mappings across timesteps; instead, the shared initialization exploits the continuity of the annealed trajectory while allowing the parameters to adapt at every stage through further optimization.
> > >
> > > The reviewer also raises an important concern regarding destabilization of earlier learned behavior during later optimization stages. In practice, this effect is mitigated through the replay-buffer mechanism.
> > >
> > > At the initial stage $t_1$, replay samples are unavailable, so we optimize only the reverse KL term by setting
> > >
> > > $$
> > > \lambda_1 = 0.
> > > $$
> > >
> > > Choosing a sufficiently small $t_1$ produces a smoother intermediate density with broadened support, making the initial optimization easier and reducing mode-collapse behavior.
> > >
> > > After learning $q_{t_1}$, transformed samples together with their log-likelihoods are stored in a persistent replay buffer. During later stages, these stored samples are reused to compute importance-weighted supervision losses for subsequent distributions. Consequently, optimization at timestep $t_{i+1}$ continually reinforces previously learned regions of the target distribution while adapting to the next annealed target.
> > >
> > > This replay-based supervision significantly reduces catastrophic drift and stabilizes sequential refinement across timesteps.
> > >
> > > Empirically, we found this progressive refinement strategy to be stable across the considered datasets/distributions.
> >
> > ### Evidence
> > 1. [Essential] Temper claims like "our method substantially improves deep learning–based sample generation" in favour of language that shows competitive performance in the settings shown.
> > > We thank the reviewer for this suggestion. We agree that the original wording was stronger than warranted by the current experimental evidence.
> > >
> > > Accordingly, we have revised statements such as “our method substantially improves deep learning–based sample generation” to more measured language that accurately reflects the scope of the presented experiments.
> > >
> > > In the revised manuscript, we  emphasize that the proposed method demonstrates competitive performance on the evaluated benchmarks relative to existing approaches.

---

> > > ### Author Response · Authors · 2026-05-21
> > >
> > > ### Evidence
> > > 2. The work would be strengthened by uncertainties on the numbers in Tables 1 and 2, describing the variance.....
> > > > We thank the reviewer for this suggestion.
> > > >
> > > > To assess the statistical significance and robustness of the reported results, we repeated all experiments using three different random seeds corresponding to independent training runs of the model.
> > > >
> > > > We now report the mean and standard deviation of all evaluation metrics across these three runs. Tables 1 and 2 have been revised accordingly to include these statistics.
> > > 3. Where data are 2D or below (c.f. MOG-40 and $\phi^4$), the work would be strengthened by visualisations of samples ....
> > > > We thank the reviewer for this suggestion.
> > > >
> > > > We have added sample visualizations for the MOG-40 dataset alongside the corresponding ground-truth distribution in the revised manuscript/appxendix. These plots help illustrate the mode coverage behavior discussed in the quantitative results.
> > > >
> > > > We did not include similar visualizations for the $\phi^4$ dataset since it is a high-dimensional distribution, making direct visualization less informative.
> > >
> > > ## Additional Comments:
> > > ### Framing and positioning
> > > 1. Can any other method optimise a non-rKL objective for sampling energy-based distributions without true samples or MCMC/Langevin?
> > > > Yes. There exist several approaches that optimize objectives other than reverse KL for sampling energy-based distributions without requiring true samples or MCMC/Langevin dynamics.
> > > >
> > > > For example, FAB optimizes the model using an $\alpha$-divergence-based objective, while iDEM employs stochastic score matching as the training objective. Both methods learn from evaluations of the target density (or energy function) and do not rely on access to true samples from the target distribution.
> > >
> > > 2. Can you explain exactly how AIS limits FAB in terms of its computational complexity, and scaling to higher dimensions? How we expect your method to compare on each of these points?
> > >
> > > > We thank the reviewer for this important question.
> > > >
> > > > In FAB, Annealed Importance Sampling (AIS) is used to improve sample quality and estimate importance weights by sequentially applying intermediate transitions between the proposal and target distributions. While effective, AIS introduces additional computational overhead because multiple intermediate distributions and transition steps must be evaluated for every sample.
> > > >
> > > > The computational cost therefore scales with:
> > > >
> > > > - the number of annealing/intermediate distributions,
> > > > - the number of transition updates per stage, and
> > > > - the dimensionality of the target distribution.
> > > >
> > > > In higher dimensions, AIS can become increasingly expensive because maintaining adequate overlap between successive distributions typically requires a larger number of intermediate annealing steps. Furthermore, transition operators may mix slowly in high-dimensional or multimodal settings, increasing the overall sampling cost.
> > > >
> > > > In contrast, our method avoids iterative AIS transitions and instead learns explicit transport mappings between successive annealed distributions. Once trained, sampling only requires a forward pass through the learned flow model, enabling direct and parallel sample generation that are stored in the replay buffer for optimising the next successive annealed distribution.
> > > >
> > > > We therefore expect our approach to offer lower sampling-time computational cost compared to AIS-based refinement methods during optimization. However, our method still relies on importance weighting between successive annealed stages, and thus does not completely avoid the general challenges associated with high-dimensional sampling.
> > > >
> > > > Overall, we expect the primary advantage of our approach relative to AIS-based methods to be reduced sequential computation during sampling and improved practical scalability through amortized transport learning (during training).

---

> > > > ### Author Response · Authors · 2026-05-21
> > > >
> > > > ### Methodology
> > > > 1. We must choose small $t_1$ to achieve connected modes in $p_{t_{1}}$, but this could make the prior very disperse compared with $p(x)$ - especially when data modes are separated by large barriers. Can this cause problems with the variance of the importance weights, since it has a long way to travel to eventually reach $p(x)$? Other factors that might affect the learnability of $q_{t_{1}}$ are (i) the fact that (without samples) the model does not know what a sensible initial state is, and may be initially very unrepresentative of $p_{t_1}$ (barely overlapping supports), and (ii) the fact that $q_{t_{1}}$  is trained with rKL, which can cause it to mode-seek over $p_{t_1}$, reducing the effective support. Do you observe any such effects in your experiments? Do you expect there to be situations in which these are problems?
> > > >
> > > > > We thank the reviewer for this insightful observation. We agree that the choice of a small initial annealing parameter $t_1$ introduces an important trade-off.
> > > > >
> > > > > On one hand, choosing a sufficiently small $t_1$ smooths the target landscape and helps connect otherwise isolated modes in
> > > > >
> > > > > $$
> > > > > p_{t_1}(x) \propto p(x)^{t_1},
> > > > > $$
> > > > >
> > > > > thereby making the initial transport problem easier. On the other hand, if $t_1$ is chosen too small, the resulting distribution may become overly diffuse relative to the final target distribution.
> > > > >
> > > > > However, the importance weights in our framework are computed only between successive annealing stages:
> > > > >
> > > > > $$
> > > > > w(x) = \frac{p_t(x)}{q_{t'}(x)},
> > > > > $$
> > > > >
> > > > > where $t'$ and $t$ are chosen sufficiently close. Thus, the prior does not directly affect the importance weights. By keeping consecutive annealing levels close, $q_{t'}$ remains a good approximation to $p_t$, ensuring adequate support overlap and stable importance weights.
> > > > >
> > > > > The reviewer also raises two important concerns regarding the learnability of $q_{t_1}$:
> > > > >
> > > > > 1. At the initial stage, no samples are available, so the model may start from a poor initialization with limited overlap with $p_{t_1}$.
> > > > >
> > > > > 2. Since $q_{t_1}$ is trained using reverse KL, mode-seeking behavior may reduce support coverage.
> > > > >
> > > > > In practice, we observed that these effects were mitigated when $t_1$ was chosen sufficiently small, since the annealed distribution becomes substantially smoother and less sharply multimodal than the original target distribution.
> > > > >
> > > > > For example, in the MOG-40 experiment, using $t_1 = 0.1$ led to noticeable mode collapse, whereas using $t_1 = 0.01$ preserved the modal structure without collapse. Although reverse KL is inherently mode-seeking, this tendency becomes less severe at early annealing stages because the softened density landscape reduces extreme probability concentration around individual modes.
> > > > >
> > > > > That said, we agree that these concerns may become significant in more challenging settings, particularly for distributions with widely separated modes and sharp energy barriers. In such cases, insufficient overlap at early stages could amplify the mode-seeking bias of reverse KL optimization.
> > > >
> > > > 2. Importance sampling becomes inefficient in high-dimensions. How does this affect the scalability of the method to high-dimensions?
> > > >
> > > > > We thank the reviewer for raising this important point. We agree that importance sampling generally becomes increasingly challenging in high-dimensional settings due to weight degeneracy and rapidly increasing variance.
> > > > >
> > > > > In our framework, however, importance sampling is not performed directly between the prior distribution and the final target distribution. Instead, the method constructs a sequence of intermediate annealed distributions
> > > > >
> > > > > $$
> > > > > p_t(x) \propto p(x)^t,
> > > > > $$
> > > > >
> > > > > and importance weights are computed only between successive stages:
> > > > >
> > > > > $$
> > > > > w(x) = \frac{p_t(x)}{q_{t'}(x)},
> > > > > \qquad t' \approx t.
> > > > > $$
> > > > >
> > > > > By ensuring that consecutive annealing levels remain sufficiently close, the overlap between $q_{t'}$ and $p_t$ is significantly improved, which helps maintain stable importance weights.
> > > > >
> > > > > Nevertheless, we acknowledge that scalability to very high-dimensional problems remains a challenging setting.
> > > >  <!-- As dimensionality increases, even small mismatches between successive distributions can accumulate and lead to higher variance in the importance weights, potentially reducing effective sample size and training stability. -->
> > > > <!-- >
> > > > > In practice, the progressive annealing strategy alleviates this issue to some extent by decomposing a difficult global transport problem into a sequence of smaller local transport steps. However, we do not claim that this fully resolves the fundamental limitations of importance sampling in high dimensions.

---

> > > > > ### Author Response · Authors · 2026-05-21
> > > > >
> > > > > ### Evidence
> > > > > 1. Abstract: "Our approach not only outperforms traditional MCMC and flow-based methods in efficiency..." What do you mean by improved efficiency in this claim, and what is the supporting evidence? Furthermore, the results do not show any comparisons with MCMC.
> > > > >
> > > > > > We thank the reviewer for pointing this out.
> > > > > >
> > > > > > By “efficiency,” we intended to refer to sampling efficiency in terms of generating high-quality samples with relatively low computational overhead once training is completed. Traditional MCMC methods generate samples through sequential stochastic transitions, where each sample depends on the previous state of the Markov chain. As a result, they often suffer from slow mixing and strong sample correlations, particularly in high-dimensional or multimodal settings. Moreover, the inherently sequential nature of MCMC sampling limits parallelization and can make large-scale sample generation computationally expensive.
> > > > > >
> > > > > > In contrast, flow-based approaches generate independent samples directly through a single forward pass after training, enabling efficient parallel sample generation and higher practical sampling throughput.
> > > > > >
> > > > > > Accordingly, we have added a runtime comparison on the $\phi^4$ $(12\times12)$ benchmark (144 dimensions), reporting sample-generation time (10000 samples) for all methods.
> > > > > >
> > > > > > | Method | Generation Time |
> > > > > > |---|---:|
> > > > > > | MCMC (HMC Algorithm) |  510 sec |
> > > > > > | FKL | 2.2 sec |
> > > > > > | RKL | 2.2 sec |
> > > > > > | FAB | 2.2 sec |
> > > > > > | iDEM | 44.0 sec |
> > > > > > | FUND (FKL) | 2.2 sec |
> > > > > > | FUND (Score) | 2.2 sec |

---

> ### Author Response · Authors · 2026-05-21
>
> **Methodology**
> 2. Figure 1 caused me quite a bit of confusion and I suggest relocating it [optional] + improving the clarity [essential]....
>
> > We thank the reviewer for carefully examining Figure 1 and for pointing out the resulting ambiguity. We agree that the current placement and presentation of the figure can be confusing.
> >
> > First, we acknowledge that placing Figure 1 near the discussion in Section 2 — which uses continuous-time notation — may incorrectly suggest that the method learns a continuous-time recursive flow. In reality, our method operates through a sequence of discrete annealing stages. To avoid this mismatch, we have relocated the figure to the Section 2.3 where the sequential modelling strategy is formally introduced. Besides we have modified the Figure 1 illustrating training as well as generation procedure clearly in the revised paper.
> >
> > We also agree that the current figure incorrectly suggests that both $z$ and $x_t$ are simultaneously provided as inputs to the same NF block. This was not the intended interpretation.
> >
> > The correct formulation is:
> >
> > $$
> > x = f_{\theta^{(t)}}(z), \qquad z \sim q_z(z),
> > $$
> >In Figure 1(A), the left panel depicts the progressive learning of intermediate annealed distributions using a unified normalizing flow (NF) model. Latent samples  $z \sim q_z(z)$ are transformed through the flow $f_{\theta^{(t)}}$ to generate samples $ x = f_{\theta^{(t)}}(z) $ corresponding to the current intermediate distribution $ q_t(x) $. The generated samples, together with their estimated log-density $ \log q_t(x)$, are reused at the next annealing stage $ t' > t $ through the replay-buffer mechanism. These replay-buffer samples are used to compute the supervision loss $ \mathcal{L}(\theta)$ via importance weighting, while fresh latent samples drawn from the prior are simultaneously used to compute the reverse KL objective. Both objectives are combined to form the final optimization objective
> >
> >$\mathcal{L_{final}}(\theta)=\mathcal{L}(\theta)+\mathcal{L_{RKL}}(\theta)$
> >
> >The dashed feedback arrows indicate iterative parameter refinement across successive annealing stages, where the same NF model is progressively updated rather than training separate flows for each intermediate distribution.
>
> >Figure 1(B) illustrates the inference or sample-generation procedure after training. Once the flow parameters are learned, independent samples can be generated efficiently by drawing latent variables $z \sim q_z(z)$ and passing them through the trained flow $f_\theta$ to obtain samples from the target distribution.

---

### Review · Reviewer_pnUU · 2026-04-23

**Summary Of Contributions:**

This paper introduces FUND, an algorithm designed to sample from unnormalised probability distributions without requiring ground truth samples. As in many modern diffusion-based generative models, the authors propose shifting the focus from sample trajectory matching to matching distribution trajectories. To achieve their goals, the authors introduce a continuouos sequence of intermediate target distributions through annealing the target density. Then, a normalising flow is trained progressively across discretized time steps to learn the evolution of the density. To avoid mode collapse, FUND uses a composite loss function that  combines a mode covering distance measure with a mode seeking reverse KL term. As the problem setup is that where true samples are unavailable, the distance measures are evaluated using an importance weighted method applied to samples drawn from a buffer. This buffer is populated by the model trained at the previously leraned intermediate step. From an architectural point of view, the authors introduce a unified modelling strategy by which a single network block is refined sequenailly across time steps, reducing memory requirements compared to incremental block building. Finally, the framework shows better performance in terms of NLL and effective sample size compared to baselines on multimodal distributions.

**Audience:**

Yes

**Audience Explanation:**

The machine learning community, particularly the researchers focused on generative models and approximate inference will for sure be interested in the findings of this paper and in the methodology behind FUND. Sampling from complex high-dimensional multimodal distributions is a fundamental challenge in ML, science and statistics. The authors herein propose a novel method to solve this problem, that has interesting properties and advantages over MCMC methods and existing sample-free flows, which tend to struggle with mode collapse.

**Broader Impact Concerns:**

Given the highly abstract nature of the work, which targets statistical mechanics, lattice field theory, and molecular dynamics, there are no immediate or direct negative societal or ethical implications.

**Claims And Evidence:**

Yes

**Claims Explanation:**

The claims made in the paper are supported by accurate and convincing empirical evidence. FUND is evaluated on a well-designed set of benchmark distributions, each testing different structural challenges. The chosen evaluation metrics (ESS, NLL, RNLL) are standard and appropriate for the problem at hand, and useful for evaluating mode collapse and sample diversity. While I am not an expert in this topic, I believe the comparisons against recent and relevant baselines are fair and well chosen, and the proposed method shows clear and robust competitive advantages. Finally, the paper includes the needed ablation and robustness studies, including variance across random initializations which makes the claims more reliable.

**Requested Changes:**

- Computational Cost Comparison: While the paper repeatedly claims improved efficiency and reduced computational overhead compared to methods like FAB and iDEM, it lacks a direct quantitative comparison of runtime. Please include a table or plot detailing the total computational cost, such as wall-clock time or total energy function evaluations.
- The details regarding minibatch construction in Algorithm 1 need clarification. During subsequent time steps, the algorithm computes importance weights for the entire buffer $B$, but the inner while loop samples fresh latent variables from $q_z(z)$. It is ambiguous how the buffer samples and their corresponding importance weights are batched and integrated with the loss calculation during the optimization step. Please clarify.
- Hyperparameter Sensitivity Analysis: The paper notes that the weights $\lambda_1$ and $\lambda_2$ are annealed based on the plateauing of the reverse NLL objective. Adding a brief ablation study or an appendix section detailing the sensitivity of the model to this specific annealing schedule would strengthen the robustness claims.

---

> ### Author Response · Authors · 2026-05-21
>
> 1. **Computational cost comparison**
> > We thank the reviewer for this valuable suggestion. We agree that direct quantitative comparisons of computational cost would strengthen the discussion regarding efficiency claims.
> >
> > Accordingly, we have added a runtime comparison on the $\phi^4$ $(12\times12)$ benchmark (144 dimensions), reporting both training time and inference/sample-generation time (10000 samples) for all methods.
> >
> > | Method | Training Time | Generation Time |
> > |---|---:|---:|
> > | MCMC (HMC Algorithm) | 0 sec | 510 sec |
> > | FKL | 0.30 hrs | 2.2 sec |
> > | RKL | 15.16 hrs | 2.2 sec |
> > | FAB | 51.20 hrs | 2.2 sec |
> > | IDEM | 5.25 hrs | 44.0 sec |
> > | FUND (FKL) | 35.69 hrs | 2.2 sec |
> > | FUND (Score) | 123.90 hrs | 2.2 sec |
> >
> > These results highlight an important trade-off:
> >
> > - methods such as FAB and FUND incur higher training cost, although for different reasons. In FUND, the additional overhead arises from progressive annealing and repeated optimization across successive intermediate distributions. In FAB, the computational cost is primarily dominated by Annealed Importance Sampling (AIS), which is repeatedly used to generate samples required for optimization.
> > - however, once trained, flow-based approaches enable extremely fast parallel sample generation through a single forward pass,
> > - whereas MCMC methods require inherently sequential sampling during inference, making large-scale sample generation computationally expensive since the runtime grows proportionally with the number of generated samples.
> > - Although IDEM requires comparatively lower training time, sample generation is substantially slower because it relies on iterative simulation of the reverse-time stochastic differential equation (SDE) using the learned diffusion score model $s_\theta$. Moreover, despite this additional inference cost, its empirical performance is generally moderate compared to FUND variants on most of the benchmarks.
> >
> > We note that although FKL exhibits very low training cost, it requires access to target samples during training and therefore does not operate in the sample-free setting considered by the rest of the methods. For this reason, we did not emphasize FKL in the computational-cost discussion, since the primary focus of the paper is learning energy-based target distributions without direct samples.
> >
> > We also note that while RKL requires comparatively moderate training time, it suffers from severe mode-collapse behavior, as reflected in the quantitative results reported in Tables 1 and 2. Consequently, lower training cost alone does not necessarily translate into better generative performance or mode coverage.
> >
> > Compared to FAB, FUND (FKL) achieves lower training cost while maintaining similarly fast inference-time generation. In contrast, Score variant of FUND incur higher training overhead due to the additional score-computation (both target as well as model score via gradient operation) in score loss optimization.
> >
> > We have included this quantitative runtime comparison in the revised manuscript and appropriately moderated our efficiency claims to distinguish between training-time and inference-time computational costs.
>
> 2.**The details regarding minibatch construction in Algorithm 1**
> > We apologize for missing this step in the algorithm. We inadvertently omitted the minibatch replay-buffer sampling step in Algorithm 1. We have corrected this in the revised manuscript and clarified how replay-buffer samples and latent samples are jointly used during optimization at subsequent annealing stages.
> >
> > Specifically, after computing and normalizing the importance weights for all samples stored in the replay buffer $B$, the optimization does not use the entire buffer at every gradient step. Instead, during each iteration of the inner optimization loop:
> >
> > - a minibatch of weighted replay-buffer samples
> >
> >   $$
> >   \{(x_b,w_b)\}_{b=1}^{m_B}
> >   $$
> >
> >   is sampled from $B$ to compute the supervision loss $\mathcal{L}(\theta)$,
> >
> > - while a fresh latent minibatch
> >
> >   $$
> >   \{z^{(1:m)}\}\sim q_z(z)
> >   $$
> >
> >   is simultaneously sampled to compute the reverse KL term $\mathcal{L}_{\mathrm{RKL}}(\theta)$.
> >
> > The final optimization objective at each minibatch step is therefore:
> >
> > $ \mathcal{L_{final}}(\theta)= \lambda_1 \mathcal{L}(\theta)+ \lambda_2 \mathcal{L_{RKL}}(\theta) $
> >
> > The model parameters are then updated using minibatch stochastic gradient descent.
> >
> > We have revised Algorithm 1 to explicitly include:
> >
> > - minibatch sampling from the replay buffer,
> > - computation of $\mathcal{L}(\theta)$ using weighted replay samples,
> > - simultaneous latent sampling for $\mathcal{L_{RKL}}(\theta)$,
> > - and the combined minibatch optimization step.
> >
> > We believe this revision resolves the ambiguity regarding minibatch construction and loss computation during subsequent annealing stages.

---

> > ### Author Response · Authors · 2026-05-21
> >
> > 3.**Hyperparameter Sensitivity Analysis**
> > > We thank the reviewer for this valuable suggestion.
> > >
> > > In the current implementation, the weighting coefficients $\lambda_1$ and $\lambda_2$ are adjusted using a heuristic annealing schedule based on the plateauing behavior of the reverse NLL objective.
> > >
> > > Within our framework, training proceeds using a composite objective
> > >
> > > $\mathcal{L_{final}}(\theta) = \lambda_1 \mathcal{L}(\theta) + \lambda_2 \mathcal{L_{RKL}}(\theta)$,
> > >
> > > where $\mathcal{L}(\theta)$ denotes either the FKL or score-matching objective, and $\mathcal{L_{RKL}}(\theta)$ denotes the reverse KL regularization term.
> > >
> > > At the first intermediate level $t=t_1$, replay-buffer samples are unavailable. Consequently, optimization initially relies only on the reverse KL term by setting
> > >
> > > $$
> > > \lambda_1 = 0, \qquad \lambda_2 = 1.
> > > $$
> > >
> > > For all subsequent timesteps, optimization additionally leverages samples generated at the previous annealing level. In this phase, $\lambda_1$ is initially assigned a relatively larger value to preserve the previously learned mode structure and maintain continuity along the annealed trajectory.
> > >
> > > This configuration is retained until improvements in the monitored reverse NLL objective begin to plateau. Beyond this point, an annealing schedule gradually decreases $\lambda_1$ while increasing $\lambda_2$, thereby transitioning the optimization toward a refinement regime in which the reverse KL term promotes sharper mode resolution and correction of residual density mismatches.
> > >
> > > We agree that understanding the sensitivity of the method to this annealing strategy is important for robustness and reproducibility. Therefore, we conducted an additional sensitivity study on the MW-8 distribution using the FUND(FKL) variant under four different annealing schedules.
> > >
> > > ---
> > >
> > > ### Annealing schedules
> > >
> > > **Setting 1**
> > >
> > > $$
> > > (1,1) \rightarrow (0.5,1) \rightarrow (0.25,1)
> > > \rightarrow (0.1,1) \rightarrow (0.01,1)
> > > $$
> > >
> > > Here, $\lambda_1$ corresponding to $\mathcal{L}(\theta)$ is gradually annealed down from $1$ to $0.01$, while $\lambda_2$ corresponding to $\mathcal{L}_{\mathrm{RKL}}(\theta)$ is kept fixed at $1$.
> > >
> > > ---
> > >
> > > **Setting 2**
> > >
> > > $$
> > > (1,0.1) \rightarrow (1,0.25)
> > > \rightarrow (1,0.5) \rightarrow (1,1)
> > > $$
> > >
> > > Here, $\lambda_1$ is fixed at $1$, while $\lambda_2$ is gradually annealed up from $0.1$ to $1$.
> > >
> > > ---
> > >
> > > **Setting 3**
> > >
> > > $$
> > > (1,0.1) \rightarrow (1,0.5)
> > > \rightarrow (0.5,1) \rightarrow (0.1,1)
> > > $$
> > >
> > > In this setting, $\lambda_1$ is annealed downward from $1$ to $0.1$, while $\lambda_2$ is annealed upward from $0.1$ to $1$ over four stages.
> > >
> > > ---
> > >
> > > **Setting 4**
> > >
> > > $$
> > > (1,0.1) \rightarrow (1,0.25)
> > > \rightarrow (1,0.5) \rightarrow (1,1)
> > > \rightarrow (0.5,1) \rightarrow (0.25,1)
> > > \rightarrow (0.1,1) \rightarrow (0.01,1)
> > > $$
> > >
> > > This schedule is similar to Setting 3, but performs the annealing more gradually over eight stages.
> > >
> > > ---
> > >
> > > In experiments, we observed that the method remains reasonably stable across a range of annealing schedules, provided that:
> > >
> > > - the $\mathcal{L}(\theta)$ term dominates during the early training phase (i.e., $\lambda_1$ remains relatively large initially), and
> > > - the transition between the two objectives occurs gradually rather than abruptly.
> > >
> > > The overall performance trends remained qualitatively consistent across all settings. However, keeping either $\lambda_1$ or $\lambda_2$ effectively fixed throughout training (as in Settings 1 and 2) was generally less favorable.
> > >
> > > Gradual annealing of both $\lambda_1$ and $\lambda_2$ produced the best empirical performance (Setting 4), suggesting that a smooth transition between mode preservation and reverse-KL-based refinement is beneficial. However, this schedule also increases the number of optimization stages and therefore incurs higher computational cost.
> > >
> > > In contrast, Setting 3 offers a favorable trade-off between computational efficiency and performance, achieving results close to Setting 4 while requiring fewer annealing stages.
> > >
> > > The quantitative comparison is summarized below:
> > >
> > > | Setting | Annealing Schedule $(\lambda_1,\lambda_2)$ | NLL ↓ | RNLL ↓ | ESS ↑ |
> > > |---|---|---:|---:|---:|
> > > | Setting 1 | $(1,1)\rightarrow(0.5,1)\rightarrow(0.25,1)\rightarrow(0.1,1)\rightarrow(0.01,1)$ | 7.07 | -33.60 | 0.79 |
> > > | Setting 2 | $(1,0.1)\rightarrow(1,0.25)\rightarrow(1,0.5)\rightarrow(1,1)$ | 7.05 | -33.46 | 0.77 |
> > > | Setting 3 | $(1,0.1)\rightarrow(1,0.5)\rightarrow(0.5,1)\rightarrow(0.1,1)$ | 7.03 | -33.62 | 0.82 |
> > > | Setting 4 | $(1,0.1)\rightarrow(1,0.25)\rightarrow(1,0.5)\rightarrow(1,1)\rightarrow(0.5,1)\rightarrow(0.25,1)\rightarrow(0.1,1)\rightarrow(0.01,1)$ | **6.96** | **-34.30** | **0.88** |
> > >
> > > we have included this sensitivity analysis and ablation study in the appendix of the revised manuscript.

---

### Comment · Action_Editor_sxcb · 2026-04-08

Dear Authors and Reviewer yDuR

We have two reviewers that are unresponsive. I will replace them by new reviewers. However, this submission is expected to be under heavy delay.

baoxiang

---

### Comment · Action_Editor_sxcb · 2026-05-10
**Extension to the author response period**

Dear Authors and Reviewers

Thank you again for submitting the manuscript and the review. The authors would like to respond to the reviews, and the authors would need some more GPU time while also taking some time for academic conferences. Therefore, we plan to provide a two-week extension to the response period.

Baoxiang

---

### Decision · Action_Editor_sxcb · 2026-06-11

**Recommendation:** Accept with minor revision

**Audience:**

Yes

**Audience Explanation:**

This paper is interesting to the sampling community

**Claims And Evidence:**

Yes

**Claims Explanation:**

All three reviewers affirm claims made in the submission supported by accurate, convincing and clear evidence, while the initial "No" votes on are based on fixable presentation issues and not method issues. With the issues fixed the manuscript is ready to be published.

The abstract still says "Our approach not only outperforms traditional MCMC and flow-based methods in efficiency and accuracy but also establishes a new method for sample-free learning of complex physical distributions." I have reservations on the *outperform* claim and would like the authors to either revise the words or further explain this to us.

---

> ### Author Response · Authors · 2026-06-24
>
> We thank the Action Editor for the positive assessment of our work and for recommending acceptance with minor revisions. We are grateful for the careful consideration of the reviewers' feedback.
> Regarding the concern about the statement in the abstract, we have revised the sentence
>
> > *"Our approach not only outperforms traditional MCMC and flow-based methods in efficiency and accuracy but also establishes a new method for sample-free learning of complex physical distributions."*
>
> to
>
> > *"The proposed approach not only improves sampling performance and accuracy over traditional MCMC and flow-based baselines but also establishes a new method for sample-free learning of complex physical distributions."*
>
> This revision avoids the stronger and more general *"outperforms"* claim while remaining faithful to the empirical results presented in the paper, which demonstrate improvements over the specific MCMC and flow-based baselines considered in our experiments.